# The MYB-related transcription factor MYPOP acts as a selective regulator of cancer cell growth

Johannes Strunk [1], Alena Hüppner[2], Mahwish Sial[1], Matthias Plath[1], Mika B. Sheriff[1], Sascha Wagner[1], Kirsten Freitag[1], Snježana Mikuličić[1], Tatjana Döring[1], Tobias Bopp [3,4,5,6], Matthias Klein[3,6], Krishnaraj Rajalingam [7], Gregory Harms[3,6,7,8], Federico Marini [6,9], Annekathrin S. Nedwed[9], Thomas Hankeln[10], Carina Osterhof [10], Marc A. Schneider [11,12], Alina Henrich[2], Andrea Nubbemeyer[2], Martin Suchan [2], Silke Brill[2], Mario Perkovic[2], Barbara Schrörs [2], Sebastian Kreiter[2], Anne Kölsch[2], Mustafa Diken [2] & Luise Florin [1,4,6] ✉

The MYB-related transcription factor and partner of profilin (MYPOP or p42POP) is a ubiquitously expressed yet understudied protein recently identified as a restriction factor of oncogenic human papillomaviruses (HPV) and proposed tumor suppressor. Here we show that in HPV-transformed cervical cancer cells, MYPOP induces alterations in cell morphology, silences viral and cellular oncogenes including *E6* and *MYC*, and stimulates the release of the cancer-killing cytokine interleukin-24. Transcriptomic and live-cell analyses reveal a rapid G1/S-phase arrest followed by loss of viable cervical cancer cells and induction of apoptosis. Moreover, MYPOP expression is broadly diminished across multiple human cancers, and its re-expression by both DNA- and mRNA-gene transfer markedly suppresses tumor cell proliferation while sparing normal epidermal keratinocytes. Similarly, murine Mypop inhibits mouse cancer cell growth. Collectively, our findings identify MYPOP as a selective suppressor of tumor cell proliferation across species in vitro and point to its potential relevance for future therapeutic investigation.

Transcription factors (TFs) play a key role in regulating cellular processes. While gene expression is orchestrated by more than a thousand different TFs in humans[1,2], the effect of individual factors differs between tissues and is sufficient to determine the fate of the entire cell[3–5]. Key TFs like MYC, MYB and p53, are critical for carcinogenesis and are dysregulated or mutated in multiple types of tumors[5–7]. Another TF which affects cell growth, migration and survival of a variety of tumor cells, but has so far attracted much less attention, is the MYB-related protein, partner of profilin MYPOP, also known as p42POP[8,9].

MYPOP was initially described as an interaction partner of the actin-binding protein profilin and was classified into the family of MYB-related TFs according to its N-terminal MYB/Saint DNA binding domain[10]. In that publication, MYPOP was shown to be able to repress transcriptional expression of the MYB-recognition element (MRE), in contrast to the inducing effect of the related proto-oncogene *MYB*[10]. In addition, MYPOP formed physical complexes with nuclear actin and histone variant H2A.Z, two regulators of transcription, chromatin remodeling and DNA repair[9,11,12].

MYPOP also appeared as an interaction partner and restriction factor of the oncogenic human papillomavirus (HPV) types 16 and 18 through binding of the minor capsid protein L2 and a region of the viral DNA[8], which is known to regulate viral oncogene expression *E6* and *E7*[13,14]. HPVs are small, non-enveloped double-stranded DNA viruses which commonly induce warts upon infection of basal skin and mucosa cells, but which are also well known as the causative agent of cervical cancer and other malignancies[15–17]. In the HPV-transformed cervical cancer cell lines HeLa, SiHa and CaSki, as well as in multiple HPV-induced tumors, the viral genome was found to be integrated into the human genome[16,18–21]. This integration contributed to the uncontrolled expression of the viral oncogenes *E6* and *E7*[19]. Interestingly, endogenous protein levels of MYPOP appeared to be strongly reduced in these virally transformed cells in comparison to healthy normal human epidermal keratinocytes (NHEK)[8], raising the possibility that MYPOP might have tumor-controlling properties. It is well known that the lack of functional tumor suppressors causes a high risk of dysregulated cell growth and, finally, the development of cancer[22,23]. Conversely, the transient re-expression of MYPOP resulted in restricted

growth across various cancer cell lines, suggesting a function of MYPOP as a suppressor of tumor cell growth[8]. This idea was supported by the observation that MYPOP reduces epithelial-mesenchymal transition, cell survival, and migration of human breast cancer cell lines (MDA-MB231, MDA-MB468, MCF-7) as well as HEK 293 T and HaCaT keratinocytes[9].

In this study, we aimed to shed light on the growth-suppressive effect of MYPOP on cancer cells by analyzing MYPOP's impact on cell morphology, cell cycle, viability and changes of gene expression. We found that MYPOP expression induced suppression of oncogenes, induction of the tumor suppressive interleukin-24 (IL-24), and a G1/S transition block in the cervical cancer cell line HeLa, as well as a significant slowdown of cell growth in both human and murine tumor cell lines, while leaving normal keratinocytes, thereby suggesting a selective effect on tumor cell proliferation.

## Results

### Induction of abnormal nuclear morphology by MYPOP
To gain deeper insights into the potential growth-suppressive role of MYPOP, we first manipulated MYPOP expression levels using plasmid DNA (pDNA) for MYPOP gene delivery and monitored its effects on tumor cells.

Initially, we performed western blot analysis to assess MYPOP protein expression levels in tumor (T) and normal (N) cells (Fig. 1a). MYPOP was abundantly detectable in normal human epidermal keratinocytes (NHEK) at 60 kDa, whereas its presence was nearly undetectable in the cervical cancer cell line HeLa, consistent with previous findings[8]. To identify specific and non-specific bands which are detected by the anti-MYPOP-antibody ab221487 (Abcam), a larger amount of protein was loaded and the resulting protein bands were compared between HeLa wild-type (WT) and MYPOP knockout (KO) samples (Supplementary Fig. 1). Band intensities revealed a substantial loss of the ~60 kDa band upon KO, corresponding to the prominent band observed in NHEK cells. Additional faint bands between 40 and 60 kDa that disappear in the knockout samples may represent less-modified, cleaved, or alternative MYPOP isoforms. In addition, nonspecific western blot bands between 68 and 130 kDa and below 35 kDa were now clearly detectable in both WT and KO cells. This indicates that the faint bands at ~100 kDa (Fig. 1a, HeLa) represent nonspecific antibody signals. Next, we expressed the MYPOP protein in the HeLa tumor cell line. Transfection with pcDNA3.1-MYPOP plasmid for untagged MYPOP and pEGFP-C3-MYPOP for GFP-tagged MYPOP resulted in the expected protein bands at approximately 60 kDa and 87 kDa, respectively (Fig. 1a, MYPOP, GFP-MYPOP). For both constructs, an additional lower-migrating band was observed, which may correspond to a less-modified form or a cleavage product. In contrast, transfection with control plasmids (pcDNA3.1, pEGFP-C3) induced either no additional bands or the anticipated GFP band at ~27 kDa.

Subsequent imaging analysis in HeLa cells revealed distinct cellular localization patterns of MYPOP. GFP alone was diffusely distributed throughout the cell, whereas GFP-MYPOP showed nuclear localization in most cells and nucleocytoplasmic and cytoplasmic distribution in some cases (Fig. 1b–e). We observed that MYPOP expression led to noticeable alterations in cell nuclei, characterized by chromatin condensation (Fig. 1c, e) shrinkage and fragmentation of the nucleus (Fig. 1g). Although GFP-transfected HeLa cells were also slightly affected, they recovered over time. Co-localization analysis using the Pearson correlation coefficient (PCC) uncovered a positive correlation, suggesting co-localization between MYPOP and the cellular DNA (PCC = +0.5), while no correlation was detected for GFP and DNA, showing independent distributions for the control (PCC = −0.2) (Fig. 1f). Counting shrunken and fragmented nuclei at 24 h and 48 h post transfection (h p.t.) demonstrated a significant increase of abnormal nuclei in cells expressing GFP-MYPOP compared to GFP (Fig. 1h, i). These nuclear changes are consistent with morphological features associated with cell cycle arrest and apoptosis[24,25].

### MYPOP induces morphological changes and cell death
To further monitor cellular alterations induced by MYPOP, we conducted flow cytometry-based assays on HeLa cells expressing either GFP or GFP-MYPOP (Fig. 2a–f). Analysis of side scatter (SSC) and forward scatter (FSC) mean values revealed significant changes not only in cell nuclei as described above, but also in cell granularity and size upon MYPOP expression, consistent with hallmarks of cell cycle arrest and programmed cell death by apoptosis[24–29]. Statistical analysis demonstrated that while cell granularity (SSC) remained unchanged at 24 h p.t., it significantly increased 48 h p.t., accompanied by a significant decrease in cell size (FSC). These morphological alterations further supported the notion of cell cycle arrest and apoptotic processes induced by MYPOP.

Employing Annexin V as a marker for apoptotic cell death and 7-AAD as a general marker for cell death (Fig. 2g–j), we explored the impact of GFP-MYPOP expression on cell viability and cell death pathways. Flow cytometry data revealed a slight reduction in double-negative cells (Annexin V-PE and 7-AAD negative, A−/7−) at 24 h p.t., which represent viable cells. However, a pronounced decrease in these viable cells was observed at 48 h p.t. Concomitantly, we detected a significant increase in Annexin V-positive (A + /7−) or 7-AAD-positive (A−/7 + ) cells at 48 h p.t. upon GFP-MYPOP expression, indicating an augmentation in apoptotic and non-apoptotic cell death. Collectively, the increase in cells stained by either Annexin V or 7-AAD (dead or dying cells) over the course of MYPOP expression suggests the activation of cell death pathways, including apoptosis, underscoring MYPOP's role in provoking cell death.

### MYPOP modulates gene expression of cytokines and cell cycle regulators
To comprehensively understand the cellular alterations induced by MYPOP, we performed RNA-sequencing (RNA-Seq) experiments comparing cells expressing GFP-MYPOP and those expressing GFP alone. In total, three independent experiments revealed 660 differentially expressed genes (DEGs) at 24 h p.t., with 182 genes significantly upregulated, and 478 genes significantly downregulated (Supplementary Data 1). The top 30 DEGs (15 upregulated, 15 downregulated) with the highest fold-change encompassed a diverse set of genes, including those encoding proteins with cytokine activity (*IL24, IL20, INHBA, TNF, IL11, IL1A*) and tumor markers (*TBC1D3D*[30], *HSPA6*[31], and *FOXD1*[32]) (Fig. 3a).

Notably, among the upregulated cytokine genes, *IL24* exhibited the most substantial increase following GFP-MYPOP expression. *IL24* encodes the well-studied tumor suppressive interleukin-24, also known as melanoma differentiation association protein-7 (MDA-7)[33–36]. In addition to the tumor marker genes, our analysis identified several well-characterized proto-oncogenes, such as *MYC, FOS*, and *JUN*, among the downregulated candidates (Supplementary Data 1). These proto-oncogenes play pivotal roles as master regulators of cell cycle entry and proliferative metabolism[37–40].

Consistent with these findings, a KEGG Pathway Enrichment Analysis revealed that the 'cell cycle' pathway ranked as the top-enriched pathway among all DEGs. Subsequent pathways included 'cancer signaling pathways,' 'apoptosis,' and 'DNA replication' (Fig. 3b). A heatmap representation of all DEGs within the 'cell cycle' pathway highlighted the expression values of significantly regulated genes, including *CDC27, TTK, PLK1, CCNB1, CCNB2, BUB1, CDC20, BUB1B, PTTG1, CCNA2, CDKN1B, MCM2-7, CDC25A, SKP2, ORC1, MYC, CDK1, E2F1, PCNA, CCNE1, CCNE2*, and *GADD45B* (Fig. 3c).

Further analysis with the STRING database of the cell cycle regulator genes, distinguishing between upregulated and downregulated genes, demonstrated how MYPOP modulates entire clusters of genes (Fig. 3d). Notably, the majority of upregulated genes were associated with the Gene Ontology (GO) term 'Negative regulation of mitotic sister chromatid separation', while 15 out of 17 downregulated genes were linked to 'G1/S transition of mitotic cell cycle'.

Given that HeLa cells are transformed by stable integration of the oncogenic human papillomavirus type 18 genome[21], we also assessed the

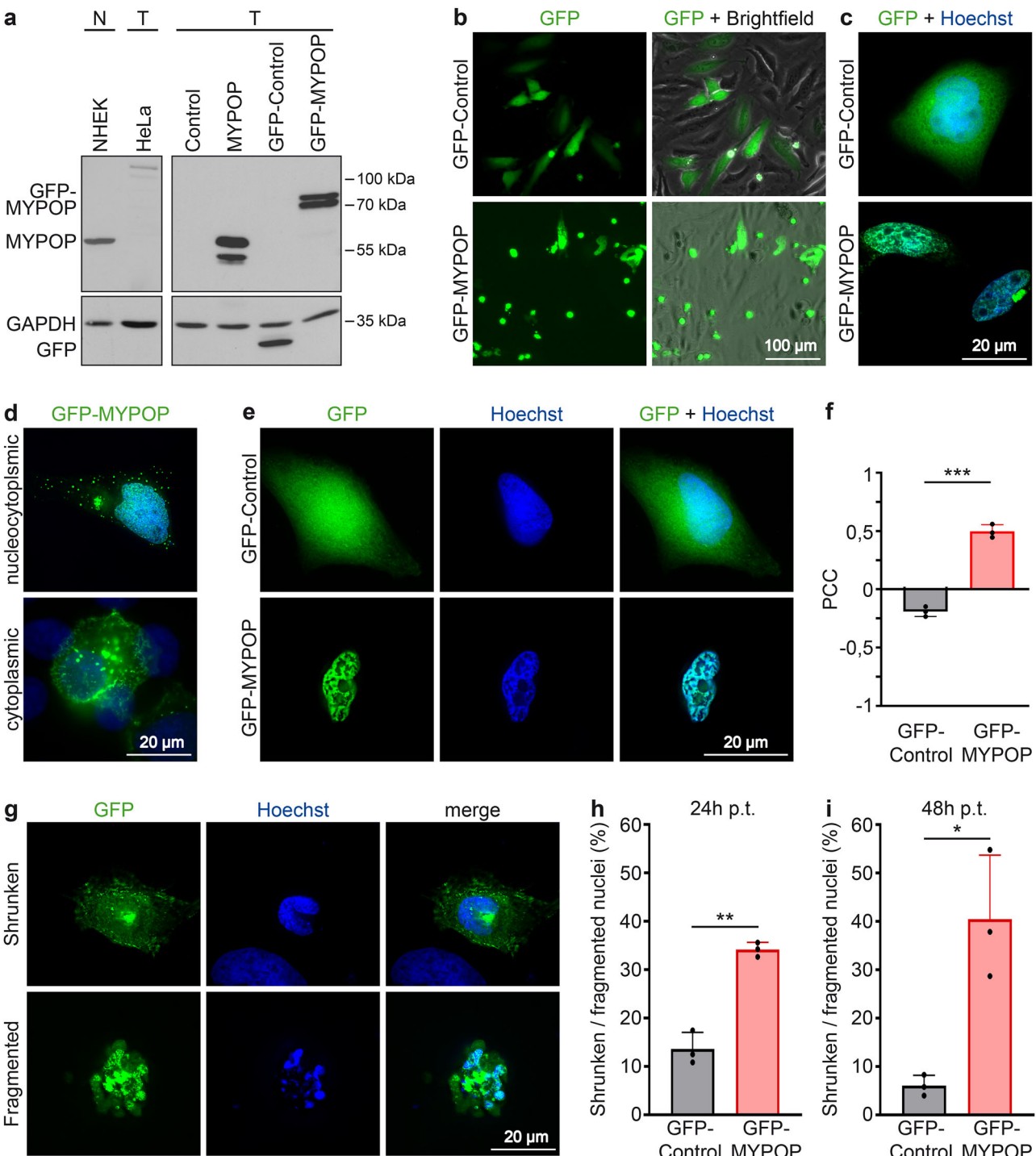

**Fig. 1 | Morphological alterations of cell nuclei in cells expressing GFP-MYPOP.**
**a** MYPOP expression in normal (N) and tumor (T) cells detected by Western blotting showing endogenous protein levels of MYPOP in normal human epidermal keratinocytes (NHEK) and cervical cancer cells (HeLa). In addition, the expression of GFP and MYPOP in HeLa cells after transfection with pcDNA3.1-MYPOP (MYPOP), pEGFP-C3-MYPOP (GFP-MYPOP) or corresponding controls (pcDNA3.1, Control and pEGFP-C3, GFP-Control) is shown. MYPOP and GFP were stained using anti-MYPOP pAb and anti-GFP mAb. GAPDH staining was used as a loading control. **b–i** GFP-Control or GFP-MYPOP expressing HeLa cells. **b**, **c** Lower (**b**) and higher (**c**) magnification for representative fluorescence microscopy images of GFP-Control or GFP-MYPOP (green) expressing cells 24 h post transfection (p.t.). Cell nuclei were stained using Hoechst 33342 (blue). **d**, **e** Representative fluorescence microscopy images treated as in (**c**) showing cytoplasmic, nucleocytoplasmic (**d**) and nuclear (**e**) localization as well as co-localization of GFP-MYPOP (green) and DNA (blue). **f** Co-localization analysis between GFP or GFP-MYPOP and DNA (Hoechst) using Pearson correlation coefficient (PCC). At least 10 GFP-positive cells were analyzed for each treatment and biological replicate ($n = 3$). Values are shown as mean + SD. Statistical significance was determined with $p = 0.0001$ comparing GFP-MYPOP and GFP-Control. **g** Representative fluorescence microscopy images of GFP-MYPOP (green) expressing cells showing shrunken or fragmented nuclei at 24 h p.t. Cell nuclei (blue) as above. **h**, **i** Quantification of fragmented and shrunken nuclei at 24 h p.t (**h**) and 48 h p.t (**i**). At least 100 GFP-positive cells were analyzed for each treatment, time point and biological replicate, respectively. Values ($n = 3$) are shown as mean + SD. Statistical significance was determined with $p$ (24 h p.t.) = 0.0036 and $p$ (48 h p.t.) = 0.0428 comparing GFP-MYPOP and GFP-Control.

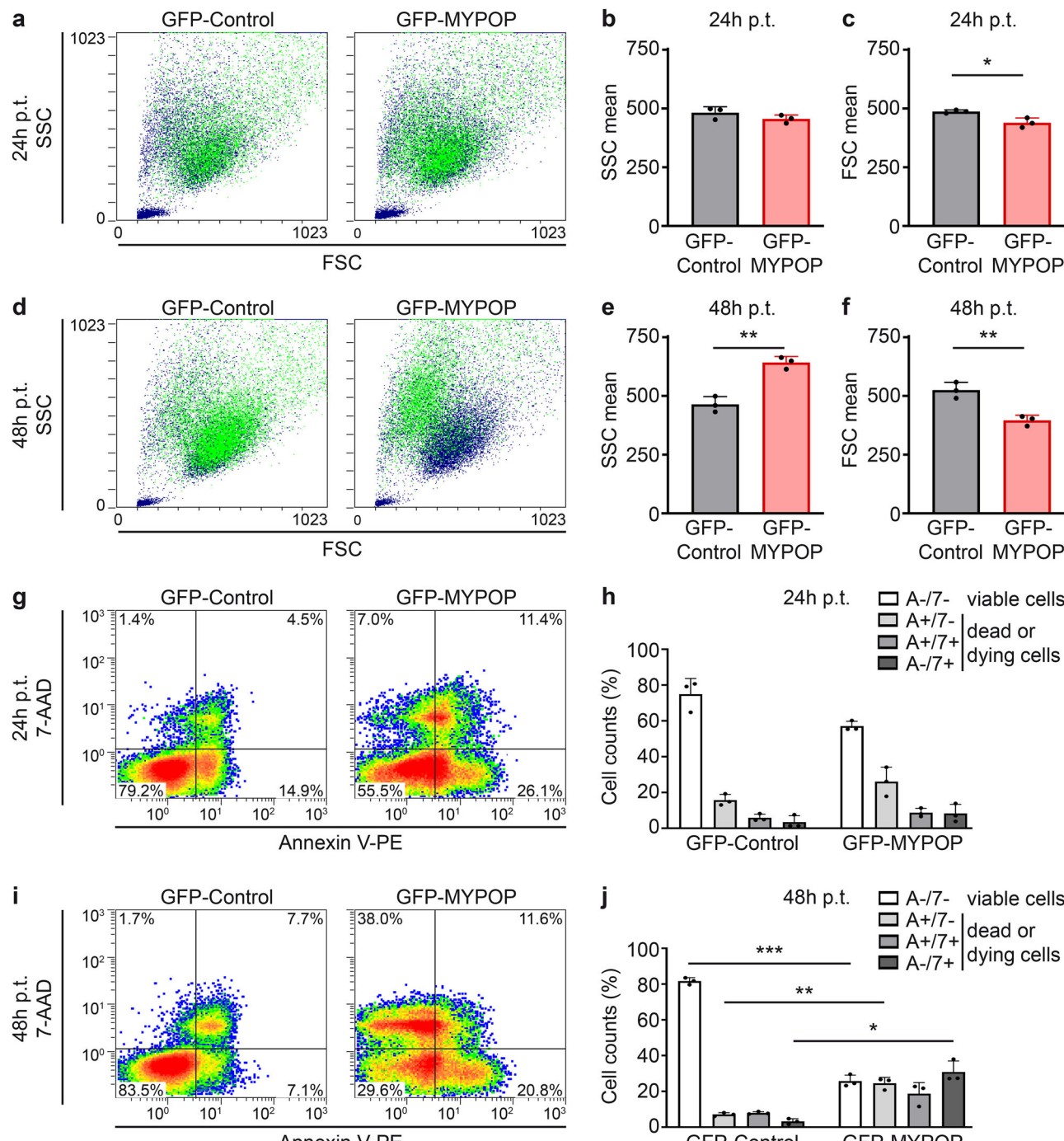

**Fig. 2 | Cell granularity, shrinkage, and cell death in GFP-MYPOP expressing HeLa cells. a, d** HeLa cells were transfected with GFP-MYPOP or the GFP-control expression plasmids as described in legend of Fig.1. SSC and FSC of the cells were determined by flow cytometry 24 h and 48 h p.t. The corresponding quantification is shown as bar diagrams displaying the SSC (**b, e**) or FSC (**c, f**) mean values ($n = 3$) + SD. Statistical significance of SSC and FSC mean values was determined between GFP-MYPOP and GFP-Control with $p = 0.2212$ (24 h p.t.) and with $p = 0.0023$ (48 h p.t.) for SSC and $p = 0.0456$ (24 h p.t.) and with $p = 0.0085$ (48 h p.t.) for FSC. **g–j** Cell death was detected using Annexin V-PE (marker for apoptotic cell death) and 7-AAD (marker for dead cells) staining of GFP-MYPOP or GFP expressing HeLa cells. Unstained cells (Annexin V negative and 7-AAD negative, A−/7−) appear in the lower left quadrant and were scored as viable cells. The lower right quadrant contains Annexin V positive and 7-AAD negative cells (A + /7−) and represents early apoptotic cells. The upper right quadrant shows double positive (A + /7 + ) late apoptotic dead cells and the upper left quadrant shows Annexin V negative/7-AAD positive cells (A−/7 + ) and generally represents dead cells. Collectively, cells stained for either Annexin or 7-AAD were also described as dead or dying cells. **h, j** show the corresponding quantification of the different cell populations at 24 h and 48 h p.t. Values ($n = 3$) are shown as mean + SD. Statistical significance was determined with $p$ (A−/7−) = 0.0609, $p$ (A + /7−) = 0.1464, $p$ (A + /7 + ) = 0.2064 and $p$ (A−/7 + ) = 0.2616 comparing GFP-MYPOP and GFP-Control (24 h p.t.) and with p (A−/7−) < 0.0001 (displayed in the figure), $p$ (A + /7−) = 0.0072, $p$ (A + /7 + ) = 0.093 and $p$ (A−/7 + ) = 0.0123 comparing GFP-MYPOP and GFP-Control (48 h p.t.).

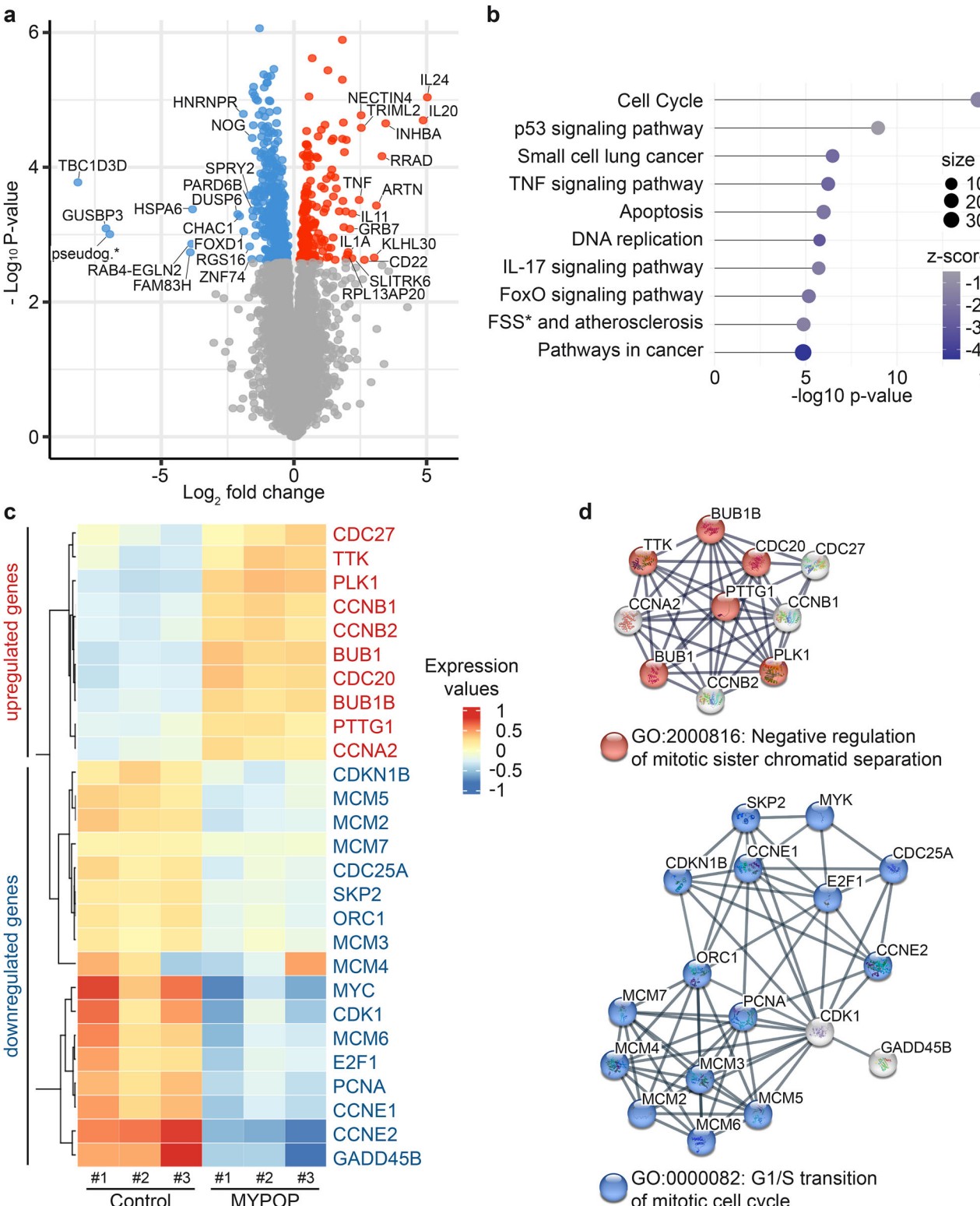

**Fig. 3 | Transcriptional response to MYPOP expression in HeLa cells. a** Volcano plot depicting gene expression changes, analyzed by RNA-Seq, comparing GFP and GFP-MYPOP plasmid transfected HeLa cells 24 h p.t. (*n* = 3). The differentially expressed genes (DEGs, adjusted *p* value ≤ 0.05) are shown in blue (downregulated) and red (upregulated). The top 15 up- and downregulated DEGs are labeled with gene names (Pseudog.* = RSL24D1 pseudogene). **b** KEGG Pathway Enrichment Analysis showing the top 10 significantly enriched KEGG pathways among all DEGs. The dot size represents the number of DEGs for each KEGG pathway. The z-score depicts the difference in the mean expression level of genes included in the respective pathway compared to the expression level of all genes included in the

analysis (*FSS = fluid shear stress). **c** Heatmap indicating the expression levels of all DEGs which were included in the KEGG pathway 'cell cycle' shown in (**b**). Each column represents one replicate (#1, #2, #3) of either GFP-Control (Control) or GFP-MYPOP (MYPOP) transfected HeLa cells. **d** STRING analysis of all DEGs which were included in the KEGG Pathway 'cell cycle' shown in c. Genes were separated into upregulated genes (with 'GO:2000816: Negative regulation of mitotic sister chromatid separation' in red) and downregulated genes (with 'GO:0000082: G1/S transition of mitotic cell cycle' in blue). Lines indicate the interaction score between proteins of the corresponding genes with the highest confidence.

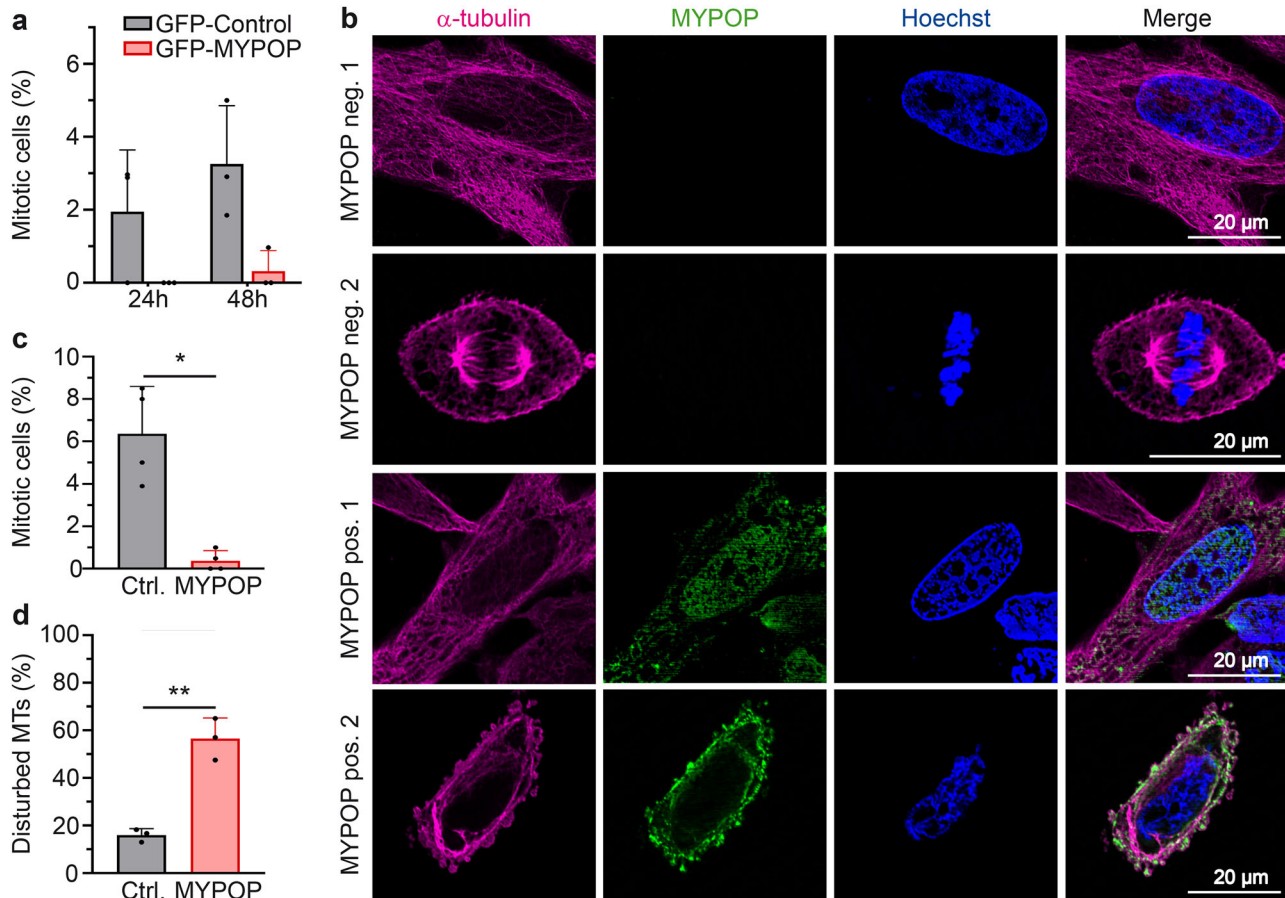

**Fig. 4 | MYPOP prevents mitosis and induces microtubule perturbation.**
**a** Mitotic cells (marked by condensed chromatin) were counted from GFP-Control
and GFP-MYPOP expressing cells by immunofluorescence microscopy. At least 100
GFP-positive HeLa cells were included in the analysis for each treatment and bio-
logical replicate at 24 h and 48 h p.t., respectively. Values ($n = 3$) are shown as mean
+ SD. Statistical significance was determined with $p = 0.1836$ (24 h p.t.) and
$p = 0.0736$ (48 h p.t.) comparing GFP-Control and GFP-MYPOP. **b** Representative
fluorescence microscopy images of non-transfected (MYPOP neg.) and MYPOP
expressing HeLa cells (MYPOP pos.) from the same samples at 24 h p.t. Micro-
tubules (MTs) were stained for anti-α-tubulin mAb (magenta). MYPOP using anti-
MYPOP pAb (green). Chromatin was stained using Hoechst 33342 (blue). Two
different images are shown for each condition (labeled as 1 and 2). **c** Mitotic cells
were counted in control (pcDNA3.1 control, Ctrl.) and MYPOP (pcDNA3.1-
MYPOP, MYPOP) transfected cells. At least 100 cells were included for each bio-
logical replicate. Values ($n = 4$) are shown as mean + SD. A statistically significant
difference was determined with $p = 0.0111$ comparing Ctrl. and MYPOP. **d** Cells
showing disturbed MTs were quantified from fluorescence microscopy assays, as
shown in (**b**). At least 100 MYPOP-positive or control-treated HeLa cells were
included for each replicate. Values ($n = 3$) are shown as mean + SD. A statistically
significant difference was determined with $p = 0.0098$ comparing Ctrl. and MYPOP.

impact of MYPOP on the expression of the viral oncogenes *E6* and *E7*. Both
of these genes are known to promote cell proliferation[41,42], and previous
HPV18 promoter analysis had suggested the silencing of HPV16 and 18
early genes by MYPOP[8]. We observed a downregulation of *E6* and *E7*
transcripts to ~75% upon GFP-MYPOP overexpression in HeLa cells after
24 h of DNA transfection; however, this reduction did not reach statistical
significance (Supplementary Fig. 2).

### MYPOP affects microtubules and prevents mitosis

To further explore the impact of GFP- and untagged, wild-type MYPOP on
mitosis, we transfected HeLa cells with GFP-MYPOP plasmid as above or
pcDNA3.1-MYPOP and the respective controls (Fig. 4a–d). Samples
expressing untagged MYPOP were stained for MYPOP, DNA, and α-
tubulin to visualize the spindle apparatus as an additional marker for mitosis
(Fig. 4b). In our single-cell, imaging-based analyses, MYPOP expression
varied widely between individual cells, ranging from very low to very high
levels. Notably, across all MYPOP-GFP–expressing cells, including those
with weak expression, we did not detect cells at 24 h p.t. undergoing mitosis,
with only a single cell showing metaphase chromosomes at 48 h p.t. in which
inhibition of sister chromatid separation could not be excluded (Fig. 4a).
Quantitative evaluation highlighted a substantial reduction in the number of
mitotic events in MYPOP-expressing cells when compared to control-
treated cells (Fig. 4a, c) suggesting that the GFP-MYPOP functions com-
parably to MYPOP without the tag as described for GFP- and Flag-tagged
MYPOP[8]. In MYPOP negative cells, we observed both the normal micro-
tubule (MT) network in non-dividing cells and the spindle in mitotic cells
(Fig. 4b, MYPOP neg.1 and 2). In contrast, a large number of MYPOP-
expressing cells displayed a disturbed microtubule (MT) network (marked
by disordered α-tubulin) (Fig. 4b, MYPOP pos. 2), which we interpreted and
scored as disturbed MTs (Fig. 4d). This finding provides a link between cell
cycle arrest and cell death as destabilization of MTs causes mitotic failure or
mitotic catastrophe, which then stimulates cell death in cancer cells[43].

### MYPOP expression reduces tumor cells growth

To extend our observation that MYPOP regulates cell growth across a
broader range of human cancers, we analyzed additional cancer cell lines
from diverse tumor origins. Western blot analysis showed that MYPOP
protein is readily detectable in normal human epidermal keratinocytes
(NHEK, as in Fig. 1a) and in normal lung cells (181576 N, 181652 N),
whereas its expression was markedly reduced in multiple cancer cell lines,
including those derived from liver (Huh7), kidney (HEK293), breast
(MCF7), colon (HCT116), cervix (HeLa; used as a control), and lung (A549,

**Fig. 5 | MYPOP pDNA transfection reduces tumor cells growth. a** Endogenous protein levels of MYPOP in untreated cancer cell lines. Normal skin cells (NHEK) and normal lung cells (181576 N, 181652 N) served as controls. GAPDH or β-actin staining was used as loading control. **b** Cancer cells were transfected with either a MYPOP expression plasmid or a control plasmid, selected for 6–12 days with G418, fixed, and stained with crystal violet. The cell-covered area was quantified (relative area). Values are presented as mean + SD. The mean for control-transfected cells was set to 100% (dotted line). Statistical significance was determined by comparing control and MYPOP expressing cells with $p = 0.0022$ for Huh7 ($n = 3$), $p = 0.0028$ for HEK293 ($n = 5$), $p = 0.0064$ for MCF7 ($n = 4$), $p = 0.0294$ for HCT116 ($n = 3$), $p = 0.0106$ for HeLa ($n = 3$), $p = 0.0003$ for A549 ($n = 3$), and $p = 0.0026$ for 2106 T ($n = 4$).

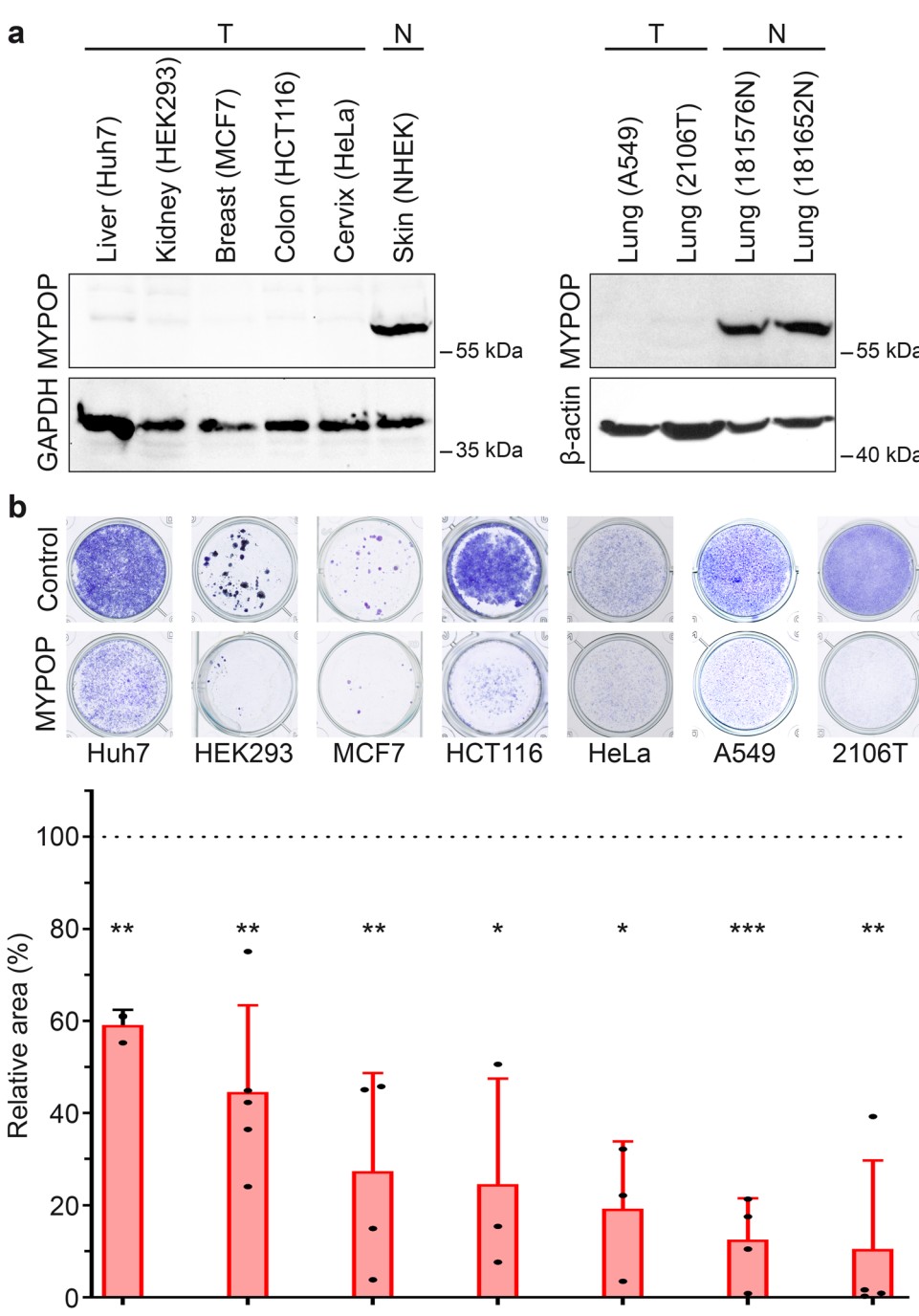

2106 T) (Fig. 5a). To assess the impact of MYPOP re-expression, we transfected these cancer cell lines with MYPOP expression plasmids, selected for transfected cells, and quantified cell numbers using crystal violet staining followed by quantitative image analysis of cell-covered area (Fig. 5b). Re-expression of MYPOP resulted in a significant decrease in the stained area, indicating robust inhibition of cell growth.

**Selective tumor cell count reduction through MYPOP mRNA**

In our pursuit of translating the knowledge about MYPOP's anti-proliferative and death-inducing properties into a potential anti-cancer strategy, we employed MYPOP mRNA transfection to express MYPOP in both tumor and normal cells using equal treatment conditions. The mRNA transfection system was selected due to its high transfection efficiencies in cell culture-based assays of tumor and primary cells and its suitability for in vivo studies and clinical trials[44–49]. A MYPOP mRNA

construct was developed for the efficient expression of MYPOP in cancer and normal cells.

A time course experiment was initiated, during which HeLa and NHEK cells were monitored for a period of 54 h, starting at 6 h p.t. to include the expected time point of peak expression[44]. The expression of MYPOP was analyzed by western blotting (Fig. 6a). Intriguingly, we observed a rapid and significant increase in MYPOP protein levels at 6 h post mRNA transfection in both HeLa and NHEK cell lysates. However, a notable difference between the cell models emerged as the MYPOP protein levels decreased to comparable levels detected for endogenous MYPOP in control-treated NHEKs (48 h and 54 h p.t.). Endogenous MYPOP was not detected in NHEK cells at 6 h p.t. due to low protein loading and short exposure times, which were optimized to prevent oversaturation of overexpressed MYPOP. In HeLa cells, mRNA-induced MYPOP expression decreased but remained elevated, with additional bands below 55 kDa likely representing unmodified or

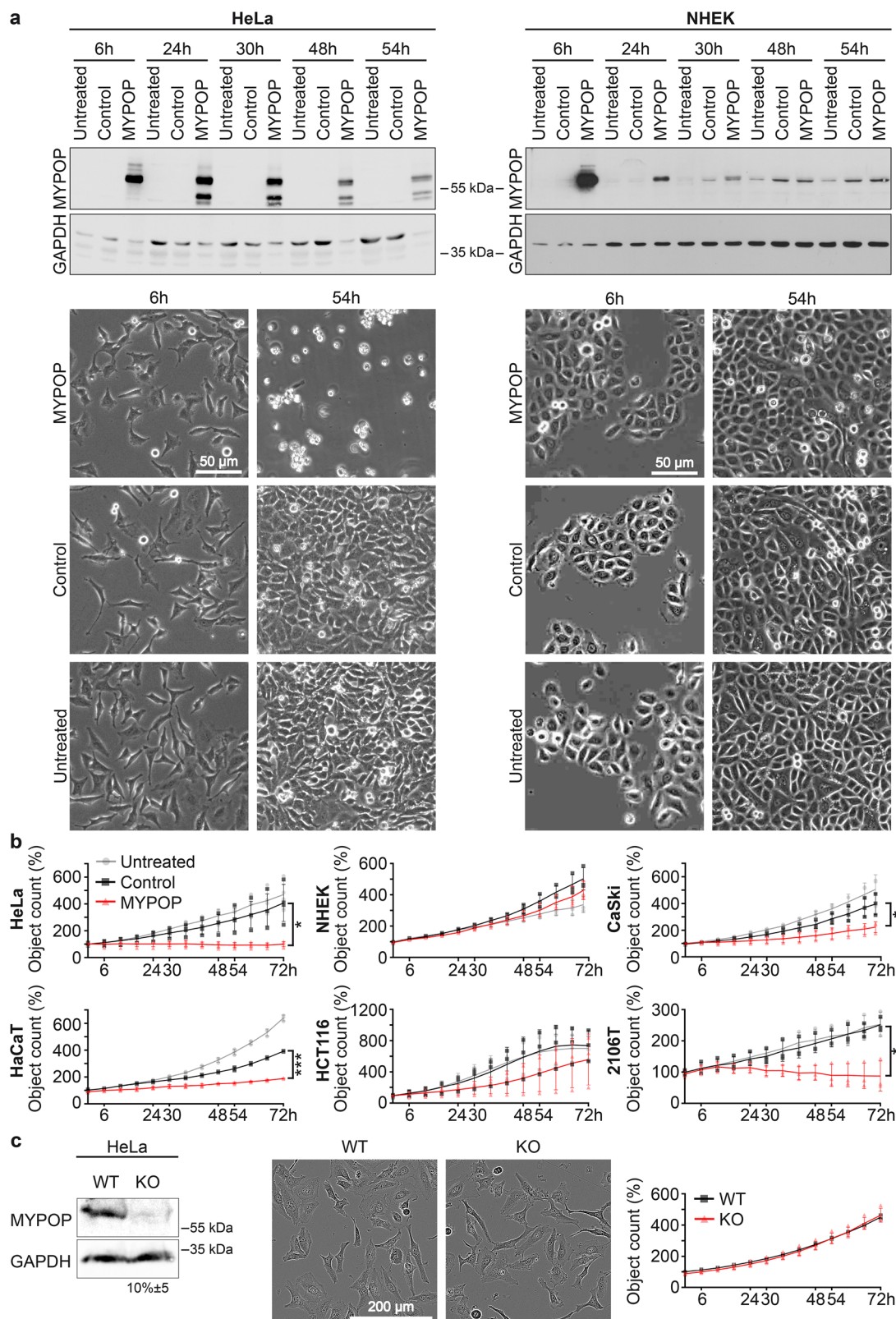

cleaved forms. Furthermore, total protein levels increased over time in all NHEK and control HeLa cells, reflecting normal cell growth as demonstrated by the loading control (Fig. 6a). In contrast, MYPOP-treated HeLa cells displayed no increase in total protein levels over time, suggesting growth arrest of these cells.

Microscopic examination of HeLa cells corroborated this observation and the expected effect of MYPOP. We noted no increase in cell confluency over time, coupled with the rounding-up of MYPOP mRNA-treated HeLa cells (Fig. 6a, Supplementary Fig. 3a). This specific morphological alteration is one of the classical features of apoptosis[50]. Conversely, NHEK cells

**Fig. 6 | MYPOP expression modulates cell growth of tumor cells. a** Protein expression of untreated, control mRNA and MYPOP mRNA-transfected HeLa cells and NHEK cells, as indicated at different time points post transfection (6 h–54 h) was analyzed by western blotting using anti-MYPOP and anti-GAPDH antibodies. Under conditions of low protein loading and short exposure times, endogenous MYPOP is not detected in NHEK, as these parameters are optimized to prevent oversaturation of overexpressed MYPOP at 6 h p.t. Detection of endogenous MYPOP increases at later time points, reflecting cell growth and higher total protein content. Lower panel: optical microscope overview images of untreated, control mRNA and MYPOP mRNA-transfected HeLa cells (left panel) and NHEK cells (right panel) at 6 h and 54 h after mRNA transfection. **b** Growth curves (object counts per image, measurement every 2 h) of untreated, control mRNA and MYPOP mRNA-transfected HeLa, NHEK, CaSki, HaCaT, HCT116, and 2106 T cells at the indicated time points. Statistical significance (*n* = 4 for HeLa, *n* = 3 for all others) was determined between Control and MYPOP cell counts at 70 h or 72 h p.t. as indicated with *p* = 0.0203 for HeLa, *p* = 0.0441 for CaSki, *p* = 0.4720 for HCT116, *p* = 0.2424 for NHEK, *p* < 0.0001 for HaCaT, and *p* = 0.0152 for 2106 T. **c** Left: MYPOP and GAPDH protein expression in Hela wild-type (WT) and MYPOP knockout (KO) cells. Statistical significance (*n* = 3) between WT and KO cells was determined with *p* = 0.0014 for MYPOP band intensities. Center: optical microscope overview images of HeLa WT and KO cells. Right: growth curve of HeLa WT and KO cells. Statistical significance (*n* = 4) was determined with *p* = 0.6099 at 72 h t.p.

displayed a normal growth behavior comparable to control and untreated cells (Fig. 6a, NHEK, Supplementary Fig. 3b).

Growth curves were generated for untreated, control mRNA-, and MYPOP mRNA-transfected cells using the Incucyte® SX5 Live-Cell Analysis System for up to 72 h p.t. (Fig. 6b). In HeLa cells, MYPOP mRNA transfection prevented an increase in cell number following MYPOP mRNA transfection throughout the observation period. In contrast, normal NHEK cells displayed growth kinetics similar to those of untreated and control mRNA-transfected cells, indicating that MYPOP expression had no adverse effect on non-transformed keratinocytes. Analysis of additional cell lines, including CaSki (HPV16-positive cervical carcinoma), HCT116, HaCaT (non-virally transformed keratinocytes), and 2106 T, further substantiated the context-dependent growth-suppressive effect of MYPOP mRNA transfection. Both CaSki and HaCaT cells exhibited a modest reduction in cell number following control mRNA transfection; however, MYPOP mRNA induced a significantly stronger and sustained growth inhibition. HCT116 cells exhibited partial growth recovery over time, whereas 2106 T cells, similar to HeLa, showed no increase in cell count during the observation period upon MYPOP expression.

Next, we examined whether eliminating the already low endogenous MYPOP levels in HeLa cells (detectable only when large amounts of protein were loaded; Fig. 6c, Supplementary Fig. 1) would enhance the transformed phenotype, as previously suggested for other tumor cells[9]. We compared the morphology and growth behavior of HeLa WT and MYPOP-KO cells (Fig. 6c) and observed no differences between the two conditions. Likewise, MYPOP knockdown in 2106 T cells using siRNA did not noticeably affect these parameters (Supplementary Fig. 4a–c).

## Selective tumor cell gene regulation by MYPOP

To investigate the reproducibility of the transcriptomic changes induced by MYPOP-pDNA as well as the selective growth-suppressive effect of MYPOP on tumor cells, we performed transcriptome analysis of HeLa and NHEK cells at 6 h and 24 h post MYPOP mRNA transfection using RNA-Seq. The DEGs are listed in Supplementary Data 1.

Comparison of the DEGs identified in HeLas after pDNA (24 h p.t.) and mRNA transfection (6 h p.t.) showed an overlap of 615 out of 660 DEGs, of those comprising a unique gene symbol, with an overlap of 158 of 182 DEGs among the upregulated candidates and 433 of 478 DEGs among the downregulated candidates (Figs. 3, and 7a). Notably, this included 25 of the previously identified top 30 DEGs (upregulated: *IL24, IL20, INHBA, RRAD, ARTN, KLHL30, CD22, TRIML2, NECTIN4, TNF, IL11, SLITRK6, GRB7, IL1A, RPL13AP20*, downregulated: *RSL24D1, HSPA6, DUSP6, CHAC1, HNRNPR, FOXD1, SPRY2, RGS16, ZNF74, NOG*) (Fig. 7, HeLa 6 h p.t.). 19 of these genes were still consistently regulated at 24 h post MYPOP mRNA treatment (Fig. 7, HeLa 24 h p.t.), including the top cytokine genes *IL24, IL20, INHBA, TNF, IL11*, and *IL1A*, irrespective of the gene delivery method or the time point of analysis. The validity of the results was also reflected by the regulation of the cell cycle genes at 6 h post mRNA treatment, including the downregulated G1/S promoting genes (*MYC, CDK1, CDKN1B, PCNA, CDC25A, SKP2, MCM2,4* and *6, CCNE1* and *2*) and upregulated mitotic checkpoint complex (MCC) genes (*CDC20, BUB1, BUB1B, CCNB1* and *PLK1*). Interestingly, while G1/S promoting genes

remained downregulated 24 h post mRNA treatment, all DEGs of the MCC shifted to the downregulated genes.

In contrast to the effect of MYPOP on the HeLa cells, we observed only a minor impact on the NHEK transcriptome (Fig. 7, NHEK). Only four of the top genes affected at 6 h post MYPOP mRNA treatment were identified (*IL24, SPRY2, ZNF74, MCM6*) (Fig. 7, NHEK 6 h p.t.), while the alterations in the transcriptome were almost undetectable at 24 h post MYPOP mRNA treatment (Fig. 7, NHEK 24 h p.t.).

These findings yield several important conclusions: first, the MYPOP-induced alterations in the cell transcriptome are consistently achieved whether MYPOP is delivered through pDNA or mRNA. Second, the N-terminal fusion of a GFP-tag does not hinder the activity of MYPOP. Third, elevating MYPOP expression has only a minor effect on the transcriptome of normal keratinocytes, explaining the absence of anti-proliferative and cytotoxic effects observed in tumor cells.

## MYPOP-mediated silencing of oncogenes and induction of cytokines

To corroborate the RNA-Seq findings, we conducted RT-qPCR analyses at 6 h and 24 h following control and MYPOP mRNA treatment (Fig. 8a). We observed that MYPOP initially repressed target oncogenes (MYC, E6, E7), but this repression faded after 24 h, even though the TFs's mRNA (Fig. 8a) and protein levels (Fig. 6a) remained stable, suggesting compensatory feedback. Simultaneously, we confirmed the upregulation of cytokine genes, specifically *IL1A, IL11, IL20*, and *IL24*.

To further validate these observations at the protein level, we employed flow cytometry-based LEGENDplex™ assays and ELISA of cell culture supernatants (Fig. 8b). These analyses demonstrated the induction and release of cytokines, such as TNF-α, IL-1α, IL-11 and IL-24, with the highest concentrations of IL-24 corresponding to the highest expression values when compared to those of the other investigated cytokine transcripts. Remarkably, the induction of these cytokines was prominently detectable 24 h p.t. and at later time points. Collectively, these results provide support for a model in which MYPOP exerts its anti-cancer effect by both silencing oncogenes and other cell cycle regulators, while simultaneously inducing the expression of the tumor-suppressive cytokine IL-24.

In addition, MYPOP mRNA rapidly blocked mitosis and disrupted microtubules, closely matching effects seen with DNA transfection (Supplementary Figs. 5 and 6). Mitotic activity was nearly eliminated by 24 h, independent of the observed expression level, accompanied by widespread microtubule disorganization. A slight recovery at 48 h p.t. suggests that MYPOP induces a transient mitotic block that recovers over time and is not strictly dependent on expression level.

## MYPOP-induced G1/S arrest

MYPOP's influence on the cell cycle and apoptosis was further investigated using HeLa cells stably expressing the FUCCI reporters hCdt1-RFP (G1 marker, red) and hGeminin-GFP2 (S/G2/M marker, green) to monitor cell cycle progression in real-time (see model in Fig. 9a). Growth curves of untreated, control, and MYPOP transfected cells indicated that marker expression did not change MYPOPs growth-suppressive effect or decrease of viable cells when compared to HeLa WT cells (Fig. 9b). For cell cycle quantification, images were analyzed using cell-by-cell analysis software

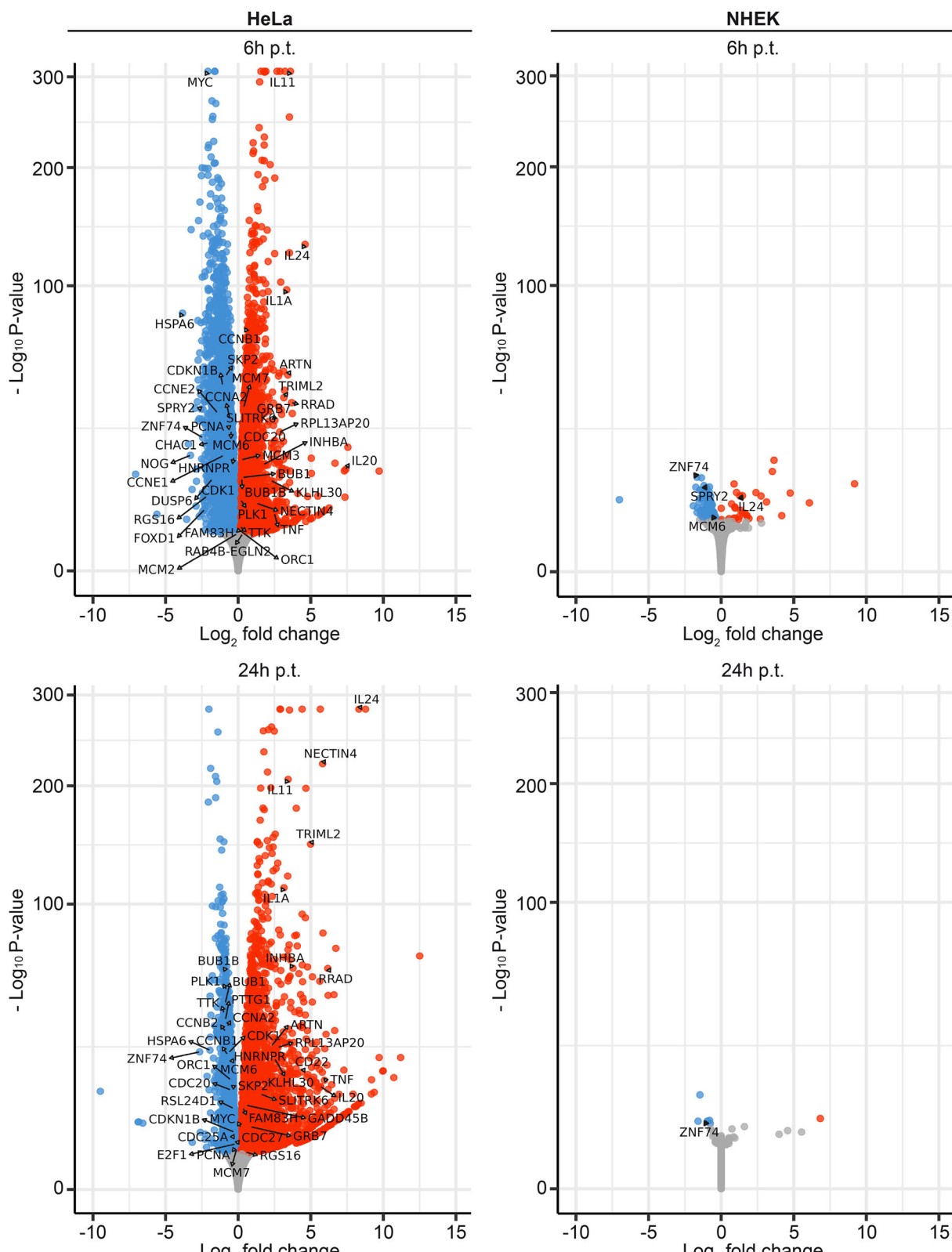

**Fig. 7 | Gene regulation in tumor (HeLa) and normal (NHEK) cells by MYPOP.** Volcano plots depicting gene expression changes of control and MYPOP mRNA-transfected HeLa and NHEK cells at 6 h and 24 h p.t. as indicated. The labels display the overlapping DEGs, which were selected from the initial RNA-Seq experiment shown in Fig. 3, comprising the 30 top DEGs (15 up, 15 downregulated) and the 27 'cell cycle' genes. Significantly downregulated candidates with adjusted $p \leq 0.05$ are shown in blue, and upregulated candidates with adjusted $p \leq 0.05$ are shown in red.

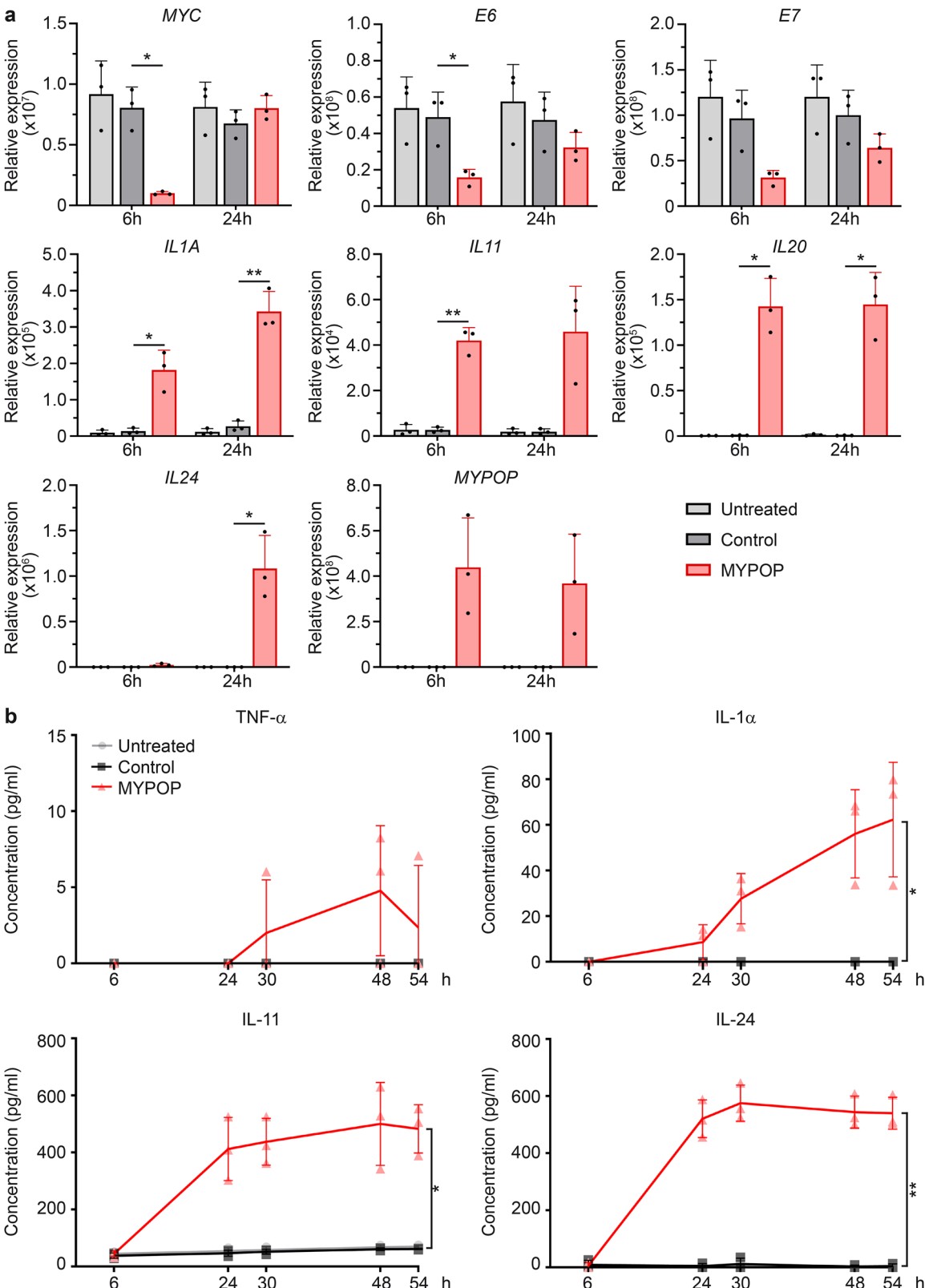

**Fig. 8 | MYPOP mRNA transfection induces the regulation of cytokines and cell cycle-regulating genes. a** Validation of DEGs from Fig. 7 in HeLa cells and *MYPOP* by RT-qPCR at 6 h and 24 h p.t. relative to GAPDH expression levels. Values (*n* = 3) are shown as mean + SD. Statistical significance was determined comparing Control and MYPOP for *MYC* with *p* (6 h) = 0.0188 and *p* (24 h) = 0.2277, for *E6* with *p* (6 h) = 0.0421 and *p* (24 h) = 0.2272, for *E7* with *p* (6 h) = 0.0609 and *p* (24 h) = 0.1397, for *IL1A* with *p* (6 h) = 0.0316 and *p* (24 h) = 0.007, for *IL11* with *p* (6 h) = 0.0053 and *p* (24 h) = 0.0616, for *IL20* with p (6 h) = 0.0154 and *p*

(24 h) = 0.0194, for *IL24* with *p* (6 h) = 0.1202 and *p* (24 h) = 0.0357 and for *MYPOP* with *p* (6 h) = 0.0731 and *p* (24 h) = 0.0995. **b** Validation of indicated MYPOP targets by ELISA and LEGENDplex™ assays of cell supernatants at time points between 6 h and 54 h post mRNA transfection. Mean values (*n* = 3) are shown as a line diagram + individual data points. Statistical significance was determined for TNF-α with *p* = 0.4226, for IL-1α with *p* = 0.05, for IL-11 with *p* = 0.0132 and for IL-24 with *p* = 0.0032 comparing control and MYPOP (54 h p.t.).

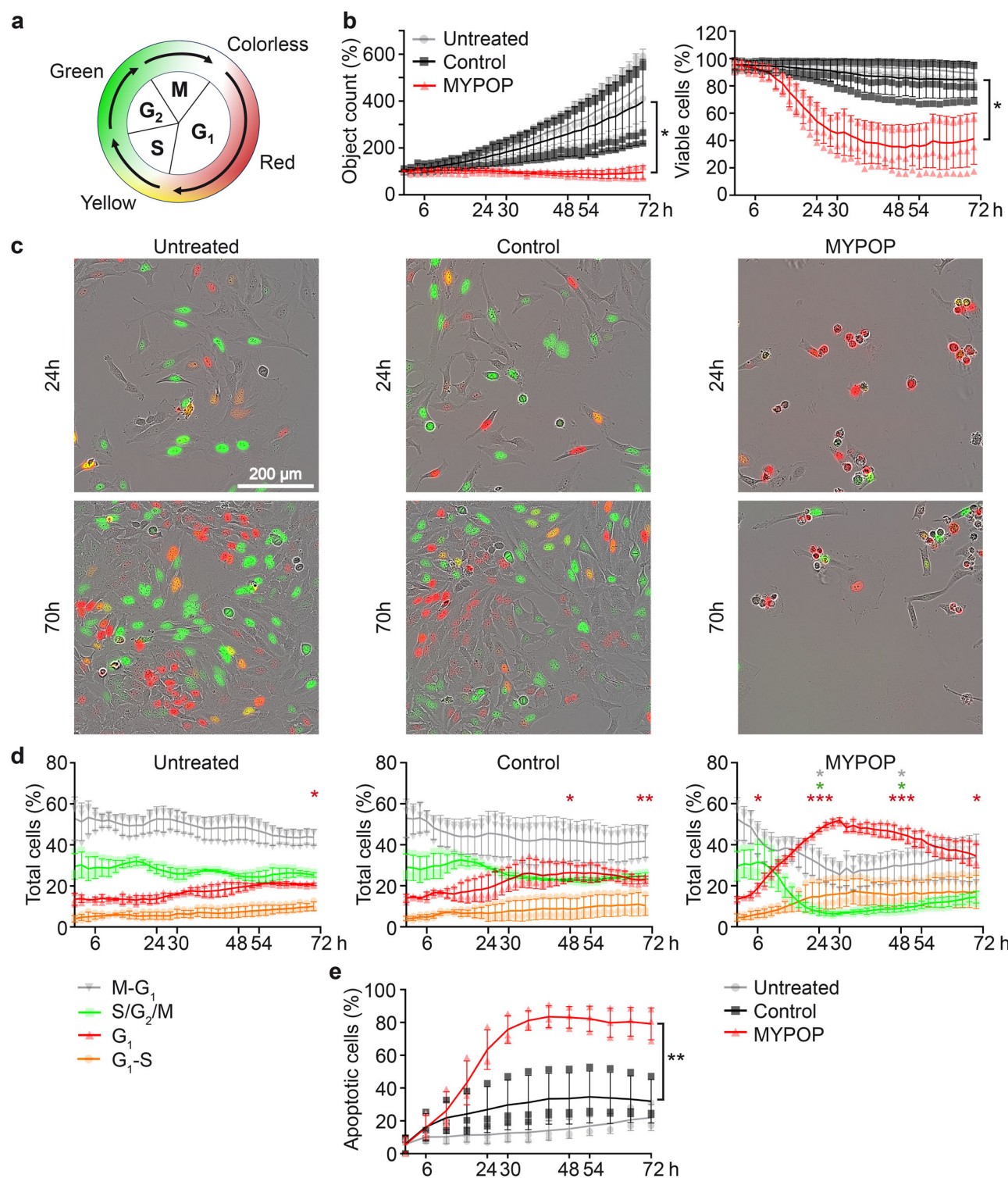

module, allowing individual cells to be classified according to colors of nuclei in G1 (red), G1-S (orange/yellow), G2 (green), and M-G1 (colorless) (Fig. 9c, d). Time-course analysis of untreated and control-transfected cells showed stable distribution of red and green fluorescence signals. In contrast, MYPOP mRNA-transfected cells displayed a pronounced increase in red fluorescence intensity, along with a marked reduction in both green and colorless cell populations immediately after mRNA transfection. This effect peaked at 24 h post-treatment, followed by a partial recovery over the next 2 days. These findings are consistent with our RNA-seq data, which revealed transcriptional signatures characteristic of G1/S arrest and downregulation of genes involved in DNA replication and cell cycle progression. Further analysis confirmed a substantial increase in annexin V-positive cells, which did not recover and remained elevated up to 72 h post-treatment (Fig. 9e).

## Murine Mypop suppresses the growth of mouse tumor cells

To investigate the murine Mypop as an anti-cancer agent in a mouse model, we employed the widely used colon carcinoma cell line CT-26 and the melanoma cell line B16-F10. Western blot analysis confirmed the efficient expression of murine Mypop (Fig. 10a). Although the reduction in cell growth mediated by MYPOP mRNA was not as potent as observed in

**Fig. 9 | MYPOP mRNA transfection induces G1/S arrest and prolonged cell death in HeLa cells. a** Schematic representation of cell cycle stages and associated fluorescent signature in Hela cells stably expressing the FUCCI reporters hCdt1-RFP (red) and hGeminin-GFP (green). **b** Growth curves and viable cell counts of untreated, control mRNA, and MYPOP mRNA-transfected HeLa cells stably expressing FUCCI reporters at the indicated time points. Statistical significance ($n = 4$) was determined between Control and MYPOP cell counts at 70 h p.t. with $p = 0.0424$ for object count and $p = 0.0113$ for viable cells. **c, d** HeLa cells treated as in b were imaged using the Incucyte live-cell imaging system. **c** Representative images of cells at 24 h and 70 h for each treatment condition. **d** Quantification of G1 (red), G1-S (orange/yellow), S/G2/M (green), and M-G1 (colorless) nuclei performed using cell-by-cell analysis software. For untreated cells, statistical significance ($n = 3$) was determined for S/G2/M with $p = 0.9123$ (6 h), $p = 0.7595$ (24 h), $p = 0.3268$

(48 h), $p = 0.3720$ (70 h), for G1 with $p = 0.9152$ (6 h), $p = 0.9974$ (24 h), $p = 0.0570$ (48 h), $p = 0.0431$ (70 h), and for M-G1 with $p = 0.9474$ (6 h), $p = 0.9747$ (24 h), $p = 0.5692$ (48 h), $p = 0.2642$ (70 h). For Control cells statistical significance ($n = 3$) was determined for S/G2/M with $p = 0.9477$ (6 h), $p = 0.9399$ (24), $p = 0.2351$ (48 h), $p = 0.3729$ (70 h), for G1 with $p = 0.4990$ (6 h), $p = 0.3017$ (24 h), $p = 0.0258$ (48 h), $p = 0.0040$ (70 h), and for M-G1 $p = 0.8231$ (6 h), $p = 0.4273$ (24 h), $p = 0.2485$ (48 h), $p = 0.1889$ (70 h). For MYPOP cells statistical significance ($n = 3$) was determined for S/G2/M with $p = 0.7672$ (6 h), $p = 0.0285$ (24 h), $p = 0.0343$ (48 h), $p = 0.0542$ (70 h), for G1 with $p = 0.0411$ (6 h), $p < 0.0001$ (24 h), $p = 0.0003$ (4 h), $p = 0.0200$ (70 h), and for M-G1 $p = 0.2927$ (6 h), $p = 0.0467$ (24 h), $p = 0.0406$ (48 h), $p = 0.1026$ (70 h). **e** Apoptotic cell analysis (Annexin V positive) of HeLas treated as in (**b**). Statistical significance ($n = 3$) was determined between Control and MYPOP cell counts at 72 h p.t. with $p = 0.0095$.

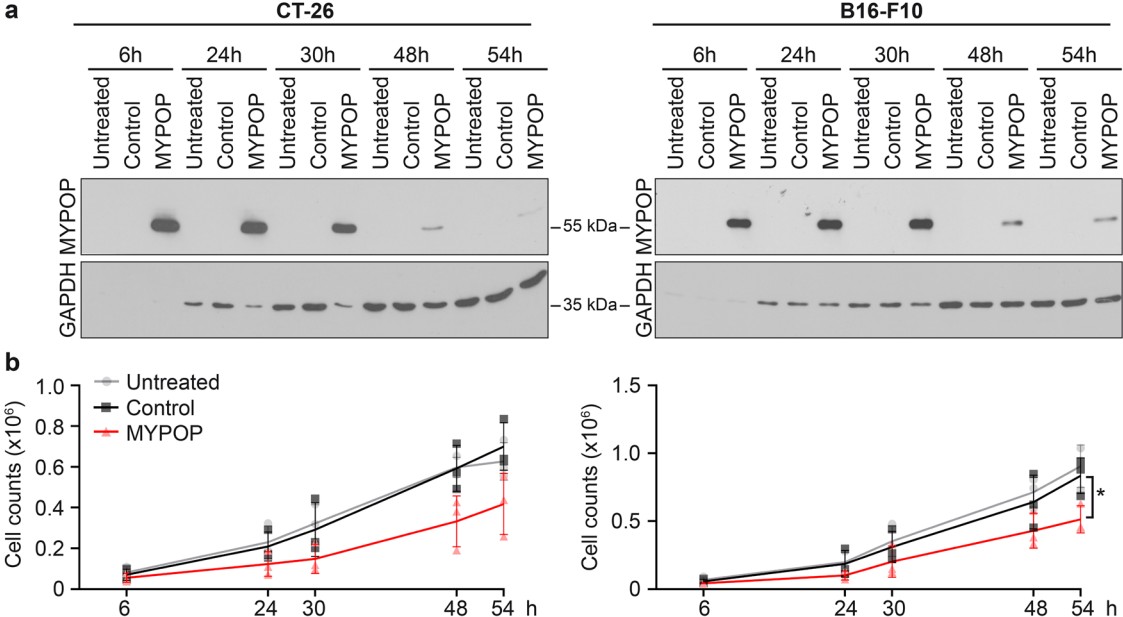

**Fig. 10 | MYPOP expression after mRNA transfection reduces cell counts in murine tumor cells. a** Protein expression of untreated, control mRNA and MYPOP mRNA-transfected CT-26 cells and B16-F10 cells at the indicated time points was analyzed by western blotting using anti-MYPOP and anti-GAPDH antibodies. **b** Cell counts of untreated, control mRNA and MYPOP mRNA-transfected CT-26 cells and B16-F10 cells at the indicated time points. Mean values ($n = 3$) are shown as a line diagram + individual data points. Statistical significance was determined comparing Control and MYPOP cell counts at 54 h p.t. with $p = 0.0650$ for CT-26 and with $p = 0.0313$ for B16-F10.

various human cancer cell lines, transfection of both CT-26 and B16-F10 cell lines with murine Mypop mRNA resulted in a 40% decrease in cell counts (Fig. 10b), suggesting the suitability of murine Mypop as a tool to suppress cancer cell growth in the mouse.

## Discussion

The results presented in this study shed light on the growth-suppressive properties of MYPOP and provide first insights into its mechanism of action. MYPOP has been studied in the context of gene delivery through pDNA and mRNA transfection, thereby demonstrating the robustness and reproducibility of the findings. We identified MYPOP as a TF whose loss is a common feature of diverse cancer cell lines. Reintroduction of MYPOP suppressed tumor cell growth in vitro in a context-dependent manner without affecting normal keratinocytes thereby suggesting a selective effect on tumor cell proliferation.

### MYPOP-induced morphological alterations and cell death

Initially, we observed a strong alteration in cell morphology induced by MYPOP expression in tumor cells. The association with condensed chromatin, the significant increase in shrunken and fragmented nuclei accompanied by cell shrinkage and an increase in granularity, is indicative of

cellular changes associated with cell cycle arrest and apoptosis. This observation aligns with the subsequent cell death analysis, which indicates induction of both apoptotic and non-apoptotic cell death pathways, further supporting MYPOP's role as a mediator of cell death in tumor cells. This is in line with previously published data showing MYPOP's role in reducing cell survival[9].

In addition, we observed a disturbance of the microtubule (MT) network in MYPOP-expressing cells. MT disruption or dysfunction of the mitotic spindle might contribute to the significant decrease in mitotic events. The study's findings are consistent with previous research, highlighting the importance of cytoskeletal dynamics and cell cycle progression in the induction of cancer cell death[43,51].

### MYPOP's impact on gene expression and cell cycle

One of the key findings of this study is MYPOPs impact on gene expression. Transient MYPOP expression caused downregulation of oncogenic targets such as MYC and HPV E6, consistent with earlier evidence for MYPOP binding to the MYB-consensus sites, which are present and functional in both the MYC and the HPV promoter[8,10,52,53]. At the moment, we can just speculate that MYPOP directly modulates transcription of these oncogenic targets. Although E6/E7 transcript levels decreased to ~30% after 6 h,

repression was not sustained despite persistent MYPOP mRNA and protein, suggesting compensatory feedback, context-dependent modulation of promoter occupancy or MYPOP modification. Detection of MYPOP protein species at ~60 kDa and lower molecular weights points to differential modification or proteolytic processing, potentially limiting the activity of MYPOP protein and thus influencing chromatin binding and transcriptional efficacy. Mapping of DNA binding sites in diverse tumor and normal cells would greatly improve our understanding of how MYPOP acts mechanistically.

The viral and cellular oncoproteins are known to promote cell proliferation through modulation of cell cycle regulators[6,41,42,54–57]. Consistent with this, we observed significant changes in the expression of multiple cell cycle-related genes. However, these effects cannot be solely attributed to secondary changes mediated by E6, E7, or MYC, since MYPOP expression transiently reduced transcript levels of these oncogenes, following MYPOP mRNA treatment, with expression recovering within a few hours. Similarly, MYPOP DNA transfection caused a mild and not significant decrease in E6 and E7 transcript levels.

Our findings rather suggest that MYPOP may directly repress a broad set of genes involved in cell cycle progression, particularly those governing the G1/S transition. Notably, these include cyclins (CCNE1, CCNE2), components of the DNA helicase machinery such as the minichromosome maintenance (MCM) complex 2–7, one subunit of the origin recognition complex (ORC), and the proliferating cell nuclear antigen (PCNA). These proteins promote DNA replication; multiple are overexpressed in various tumors and are therefore promising targets in cancer therapy[58–62]. Likewise, abrogation or prolonged activation of the mitotic checkpoint, as observed post MYPOP expression, provokes defective replication and apoptosis in cancer cells[63–65]. More precisely, mitotic failure caused by defective replication leads to mitotic catastrophe and eventually to cancer cell death[43,66,67].

In line with this, live-cell imaging combined with cell cycle, viability and apoptosis assays revealed a pronounced block at the G1/S transition and a subsequent decrease in viable cells and sustained increase in apoptotic cells following MYPOP mRNA treatment. The G1/S arrest appeared to be alleviated approximately three days post-treatment, likely reflecting the short half-life of the mRNA and the consequent reduction of MYPOP protein levels over time. These observations imply that for potential therapeutic applications in cancer, repeated MYPOP mRNA administration may be required to maintain sufficient protein levels, or alternatively, vector-based DNA gene therapy could provide a more sustained expression profile.

In addition, MYPOP activates cytokine expression and secretion, including TNF-α, IL-1α, IL-11 and IL-24. These cytokines play crucial roles in wound healing processes and in the tumor microenvironment (TME)[68,69]. Hereby, of particular interest is the tumor suppressive cytokine IL-24, a member of the IL-10 family also called MDA-7, which among other functions in cell differentiation, host defense, and wound healing[70,71] was shown to induce apoptosis in tumor cells, as demonstrated in a phase I clinical trial[33,36,72]. This implies that MYPOP causes multiple (primary and secondary) effects, which might collectively contribute to the inhibition of cell growth and subsequent cell death.

### Selective tumor growth suppression by MYPOP
MYPOP mRNA transfection facilitated efficient MYPOP expression in both tumor and primary normal cells. This resulted in a marked reduction in cell numbers in most tested human and mouse tumor cell lines, a transient growth inhibition in cells such as HCT116, and undetectable effects on normal keratinocytes, indicating that MYPOP overexpression acts in a context-dependent manner rather than as a nonspecific cytotoxic factor.

This is in line with the expression value data: enhancing MYPOP protein to supra-physiological levels in normal keratinocytes caused only minor alterations in the transcriptome, indicated by both the amount of DEGs and the respective level of regulation. This differential responsiveness may arise from cell-type-specific differences in chromatin landscape or cofactor availability at MYPOP target promoters and warrants further investigation, including validation across more normal primary cell types.

Previous studies have shown that cooperative or antagonistic interactions among TFs can profoundly influence their occupancy at regulatory DNA elements[4]. Thus, delineating these TF interactions will be critical for understanding the context-dependent functions of MYPOP. In this scenario, the endogenous protein levels of MYPOP might play a role. Endogenous MYPOP is readily detectable in primary keratinocytes but strongly reduced in cancer cells, as shown here and previously[8], suggesting that normal cells are intrinsically adapted to tolerate MYPOP activity, whereas tumor cells have undergone selection to downregulate it and thus become dependent on its loss. Consequently, re-expression of MYPOP in tumor cells may unmask this acquired vulnerability and elicits anti-proliferative and pro-apoptotic responses, while even supra-physiological MYPOP levels in NHEKs cause only minor transcriptomic and phenotypic changes.

Intriguingly, this selective induction of cell death has also been observed for the previously mentioned cytokine IL-24[36,70,73,74], which is induced by MYPOP and underscores the clinical potential of MYPOP as an anti-cancer protein. In addition, MYPOP-gene delivery exerted anti-proliferative effects in both human and mouse cancer cells. This finding suggests that MYPOP's function may be generalizable across species, adding to its potential applicability for pre-clinical studies.

Interestingly, although Du et al. convincingly reported increased migration and survival after MYPOP siRNA treatment of HEK293T, HaCaT, and MCF-7 cells, our finding that MYPOP depletion does not affect HeLa or 2106 T morphology or growth indicates that MYPOP's influence is context-dependent, as reported for multiple TFs[4]. These findings also suggest that MYPOP expression in HeLa and 2106 T cells is already repressed below the threshold necessary for growth control, and that further reduction does not confer any additional proliferative advantage as reported for various tumor suppressors[75].

The preferential effect of MYPOP re-expression on tumor cells remains unresolved. As our and the above-mentioned study are based on MYPOP overexpression, or loss-of-function analyses in tumor cells, efficient MYPOP knockout analyses in primary untransformed cells will be important to determine whether MYPOP acts as a bona fide tumor suppressor under physiological conditions.

### Concluding remarks and outlook
This study provides exploratory observations on the effects of MYPOP in cervical cancer cells. Alterations in gene expression detected 6 h after MYPOP mRNA treatment, followed by cytokine release and changes in microtubule integrity at 24 h, are consistent with a temporally separated cellular response. Early transcriptional changes may include reduced expression of some genes associated with G1/S progression. These changes could be associated with later responses such as IL-24 induction, which may contribute to the limited growth-inhibitory effects observed. IL-24 secretion may also be linked to paracrine signaling, although its functional relevance remains uncertain. Additional studies are needed to identify MYPOP's direct genomic targets and to explain its context-dependent effects on the transcriptome and cell growth, as well as the link between early and later effects.

Collectively, these findings suggest that MYPOP-based gene delivery may have transient and context-dependent anti-tumor effects. Whether sustained delivery or combination treatments can meaningfully enhance its activity, similar to other G1/S-blocking agents[76], remains to be determined. Substantial preclinical in vivo studies are still needed to assess its safety and overall therapeutic potential.

## Methods
### Cell culture
The human cervical carcinoma cell line (HeLa) that contains multiple copies of integrated HPV 18 DNA[21], was purchased from the German Resource Center of Biological Material (DSMZ, Germany). Cells were cultured in Dulbecco's Modified Eagle Medium (DMEM, Thermo Fisher Scientific, USA) supplemented with 10% fetal bovine serum (FBS, Merck KGaA, Germany) or in Minimum Essential Medium Eagle (EMEM, HiMedia,

Germany) supplemented with 10% heat-inactivated FBS (SUPERIOR stabil, Bio&Sell, Germany). Human hepatocellular carcinoma cell line Huh7 was provided by Dr. Reinhild Prange, human breast adenocarcinoma cell line MCF7 by Dr. Walburgis Brenner, colorectal cell line HCT116 by Dr. Thomas Kindler, (all University Medical Center, Mainz, Germany), and human embryonic kidney 293 (HEK293) cells were obtained from ATCC (CRL-1573). Huh7, MCF7, and HEK293 cells were cultured in DMEM supplemented with 10% FBS and 5 µg/mL ciprofloxacin (Fresenius Kabi, Bad Homburg, Germany). HCT116 were cultured in McCoy´s 5 A medium supplemented with 10% FCS and 1% penicillin-streptomycin. The lung adenocarcinoma cell line A549 was purchased from ATCC (#CCL-185, USA). The lung cell line 2106 T, as well as primary normal lung cells 181576 N and 181652 N were provided by the Lung Biobank Heidelberg, member of the NCT tissue bank, the Biomaterial Bank Heidelberg (BMBH) and the platform biobanking of the German Center of Lung Research (DZL). The squamous cell carcinoma cell line 2106 T has been intensively characterized[75]. A549 and 2106 T were cultured in DMEM/Ham´s F-12 medium supplemented with 10% FCS. 181576 N and 181652 N normal lung cells were derived from a lobectomy. Lung tissue was minced and dissociated with Liberase H (ROCHE, Germany) for 2 h at 37 °C on a wheel and cultured in 2D in DMEM/Ham´s F-12 medium supplemented with the PromoCell Airway Epithelial Cell Growth Medium Supplement Pack (PromoCell, Germany) and ROCK-inhibitor (Stemcell Technologies, Germany). Murine B16-F10 cells (ATCC, CRL-6475) were kept in DMEM (Thermo Fisher Scientific, USA) supplemented with 10% fetal bovine serum (FBS SUPERIOR stabil, Bio&Sell, Germany). The murine CT-26 cell line (ATCC, CRL-2638) was cultured in RPMI Medium 1640 (Thermo Fisher Scientific, USA) supplemented with 10% FBS (SUPERIOR stabil, Bio&Sell). All cells were cultured at 37 °C in a humidified environment containing 5% CO2, except for B16-F10 cells, which were maintained at 7.5% $CO_2$.

Bacterial or fungal contamination was monitored through daily visual cell microscope-based monitoring. Exclusion of mycoplasma contamination and reauthentication of cell lines were performed by PCR and short tandem repeat (STR) profiling, respectively, at Eurofins, Germany. No commonly misidentified cell lines were used. Normal Human Epidermal Keratinocytes (NHEK) were purchased from PromoCell, Germany and were cultivated according to the manufacturer's instructions.

### Antibodies
Antibodies used in this study were as follows: anti-MYPOP rabbit polyclonal antibody ab221487 (Abcam, UK); anti-GFP mouse monoclonal antibody (mAb) JL-8 (Clontech, USA); anti-GAPDH mAb 60004-1-Ig (Proteintech, Germany); anti-α-tubulin mAb T5168 and anti-β-actin mAb A5441 (Merck KGaA, Germany); anti-mouse and anti-rabbit horseradish peroxidase-coupled secondary antibodies (Dianova, Germany), Alexa Fluor 488- or Alexa Fluor 546-coupled secondary antibodies (Thermo Fisher Scientific, USA).

### Transfection
For pDNA-mediated MYPOP gene transfer, cells were seeded into cell culture dishes to a confluency of 70–90%. Transfection was carried out using polyethylenimine (PEI, Merck KGaA, Germany) or Lipofectamine 2000 (Thermo Fisher Scientific, USA) together with expression plasmids encoding one of the following constructs: pEGFP-C3-MYPOP, p3xFLAG-CMV-10 MYPOP, or an untagged MYPOP pcDNA3.1( + )-MYPOP[8]. Corresponding empty expression plasmids pEGFP-C3 (Clontech, USA), pcDNA3.1(+) (Invitrogen, USA), and p3xFLAG-CMV-10 (Sigma Aldrich) were used as controls.

For preparation of mRNA constructs, synthetic DNA fragments coding for the human MYPOP (accession number NM_001012643) or murine MYPOP protein (accession number NM_145579), as well as firefly luciferase and EGFP were cloned into a plasmid vector derived from pSTI vector[77,78]. Nucleoside-modified mRNAs, were synthetized using triphosphate derivative N1-Methylpseudouridine-5'-Triphosphate in the

transcription reaction, and double-stranded mRNA contaminants were removed via cellulose purification as described before[79,80]. RNA concentration and purity were assessed by spectrophotometry (NanoDrop 2000c, Thermo Fisher Scientific, USA). RNA integrity of synthetic RNA was assessed by capillary electrophoresis (Fragment Analyzer; Agilent Technologies, USA).

For mRNA-mediated MYPOP gene transfer, cells were harvested after detaching with Accutase® (HiMedia, Germany) and cell counts were determined using erythrosine B (Merck KGaA, Germany). Cells were transfected 24 h after seeding ($1 \times 10E4/cm^2$) with the respective mRNAs (0.813 µg/well) using Lipofectamine MessengerMAX mRNA Transfection Reagent (LMRNA, Thermo Fisher Scientific, USA) according to the manufacturer's protocol. Phase-contrast images were acquired at each time point.

### Western blot
Protein expression levels were confirmed by Western blotting. Cells were resuspended in Laemmli sample buffer supplemented with beta-mercaptoethanol (β-ME, final concentration of 10% (v/v)). Lysates were then sonicated three times for 20 seconds at 4 °C using a Bioruptor® Plus Sonication Device (Diagenode, Belgium) and subsequently heated for 10 min at 95 °C. Lysates were loaded onto a 10% SDS-PAGE for protein segregation and plotted onto a nitrocellulose membrane (Cytiva, Amersham, USA). Unspecific antibody binding sites were blocked with PBS containing 5% nonfat dried milk (w/v, AppliChem, Germany), and primary antibodies were incubated at 4 °C overnight. Following the incubation of primary antibodies, horseradish peroxidase-coupled secondary antibodies were incubated 1 h at room temperature. Protein bands were visualized using substrate for enhanced chemiluminescence (ECL, PerkinElmer, USA). Densitometric analysis was performed using ImageJ 1.54 f software (Wayne Rasband and contributors, National Institutes of Health, USA, http://imagej.org). Uncropped blots/gels are shown in Supplementary Figs. 7–12.

### Immunofluorescence microscopy
For immunofluorescence images, HeLa cells were cultured on glass coverslips and transfected as described above. Cells were fixed for 10 min in 4% paraformaldehyde (PFA) in Dulbecco's Phosphate-Buffered Saline (DPBS, Thermo Fisher Scientific, USA) and permeabilized in 0.2% TritonX100 in DPBS at room temperature. Unspecific antibody binding sites were blocked with 1% bovine serum albumin (BSA, AppliChem, Germany) in DPBS (w/v). Primary and secondary antibodies were incubated 1 h at room temperature, respectively. Nuclei (DNA) were stained using Hoechst 33342 (Thermo Fisher Scientific, USA).

Fluorescence imaging was conducted using an Axiovert 200 M fluorescence microscope. Z-stack images were captured and deconvoluted using the software supplied by Zeiss (Axiovision 4.7, Carl Zeiss, Germany). For higher resolution images, fluorescence imaging was conducted using a Confocal Laser Scanning Microscope (TCS SP8, Leica, Germany) with a 63 × 1.4 NA oil immersion objective. The images were always acquired at a factor of at least 2.3 times less than the calculated confocal lateral and axial resolution, at a scan rate of 400 lines/minute and with 2× averaging. All images used in comparison were prepared and acquired under the same conditions. Control images acquired of just single stained cells or of unstained cells revealed that under these detection and imaging schemes and under the conditions for imaging above, cross-talk fluorescence, background fluorescence and background autofluorescence were either non-existent or so low that they would be insignificant. The images shown in the figures were smoothed with the standard Leica smoothing algorithm. In some cases, some signals had up to a 20% cutoff and/or a 20% threshold applied in order to remove background fluorescence and/or cellular autofluorescence and to create homogeneous cell borders with the LASX Software (Leica, Germany) and were deconvoluted with the Leica Lightning Application within the LASX Software (Leica Mannheim, Germany) with combinations of nearest neighbor and Wiener filter algorithms.

For co-localization analysis, HeLa cells expressing GFP or GFP-MYPOP were stained for DNA using Hoechst. Cells showing GFP or nuclear GFP-MYPOP localization were selected for image acquisition using a Zeiss Axiovert 200 M microscope fitted with a Plan-Apochromat 100 Å~/ 1.4 Oil objective (Carl Zeiss, Jena, Germany). Quantification of co-localization was performed by analysis of at least 10 pictures per group using the Pearson correlation coefficient (Co-localization Software 4.7, Carl Zeiss).

## Generation of knockout cell lines HELA_MYPOP-KO

The CRISPR-Cas9 system was used for the knockout of MYPOP (myPOP) (Gene ID: 339344). Cocktail of three modified sgRNA targeting human myPOP (Gene Knockout Kit v2, Synthego, USA; guide sequences: 5'-CACGCGAGCGAGCUUCUCCU-3', 5'-CUGGAAGCGCACGGGC-CAGG-3', 5'-GCCCACUACCCGCAGCUCUA-3') and Cas9 (CleanCap Cas9 mRNA, L-7606-100, TriLink, USA) encoding mRNA was electro-porated into HELA cells (three square wave pulses of 90 V/mm, each 6 msec. and 400 msec interval at RT). Cells were screened via Sanger sequencing of genomic DNA (Eurofins, Germany), amplicons generated by PCR (primer sequences: 5'- AAACCACCCGGTTGCGCAAG-3' and 5'-CAGCC-TACTCTGAAGACACTC-3'). Cells were tested in vitro, expanded, and cell banks were generated for further experiments.

## Cell death assay

HeLa cells were cultivated and transfected as described above using GFP-control and GFP-MYPOP expression plasmids. Cells were collected 24 h or 48 h p.t. and were prepared for flow cytometry according to the manu-facturer's protocol using the eBioscience™ Annexin V Apoptosis Detection Kit (Thermo Fisher Scientific, USA). In brief, 5x10E5 cells were washed in PBS and Annexin Binding buffer. Cells were then resuspended in Annexin Binding buffer and stained with Annexin V-PE and 7-AAD. Cytofluoro-metric analysis was performed with flow cytometer FC500 and CXP analysis software (Beckman Coulter, USA). Thereby, GFP-positive cells were ana-lyzed regarding their staining for Annexin V and 7-AAD. Annexin V-negative and 7-AAD-negative cells (A−/7−) were considered as viable cells. Annexin V-positive and 7-AAD negative cells (A + /7−) were scored as early apoptotic cells. Double positive cells (A + /7 + ) were considered as late apoptotic or non-apoptotic dead cells, and Annexin V negative/7-AAD positive cells (A−/7 + ) as non-apoptotic dead cells. Cells showing either A + or 7+ were also collectively considered dead or dying cells. Gating strategy is shown in Supplementary Fig. 13. Additionally, FSC and side scatter (SSC) were used to illustrate the influence of MYPOP regarding cell size and granularity.

Cell death assays were also performed using mRNA constructs for the transfection of MYPOP. Untreated HeLa cells and mRNA coding for luciferase served as controls. HeLa cells were stained with Annexin V-FITC (1:100) (ImmunoTools, Germany) and 7'AAD (1:100) (Merck KGaA, Germany) 24 h p.t. Briefly, $1 \times 10E6$ cells were stained with Annexin V-FITC and 7'AAD for 10 min at 4 °C protected from light. Fluorescence was measured using the BD LSR Fortessa Cell Analyzer followed by analysis of cell apoptosis using FlowJo v10.8.1 software.

## Crystal violet staining

The effect of MYPOP on the growth of different cell types was determined using crystal violet staining. Cells were transfected with pEGFP-C3-MYPOP, p3xFLAG-CMV-10 MYPOP or the corresponding control. Subsequently, cells were cultured in growth medium containing neomycin (G418) to select for transfected cells. The medium was replaced every two to three days for ~6–12 days, depending on the cell type. At the end of the incubation period, cells were washed with DPBS, fixed with 99.9% methanol, and stained with crystal violet. Finally, the area covered by cells was quantified using the 'Colony Area' plugin with ImageJ (v1.53.t[81]).

## Growth Curve generation of different cell lines transfected with human MYPOP mRNA via Incucyte®

Cells of interest were transfected with human MYPOP-mRNA or either control-mRNA 24 h after seeding into 12-well plates as described above. All samples were prepared in quadruplicate. Following Transfection, the 12-well plate was placed in Incucyte® Live Cell Analysis System for visual monitoring for 72 h. Measurements and microscopic images per well were taken every two hours. Incucyte® AI Cell Health Analysis Software Module was used for analysis. This Module is an all-in-one analysis tool that segments cells (Object counts) and classifies them as live or dead.

## Reverse transcription quantitative polymerase chain reaction

Gene expression of target genes was determined via Reverse transcription quantitative polymerase chain reaction (RT-qPCR). Total RNA was extracted from the cell line sample material using QIAGEN RNeasy Micro Kit according to the manufacturer's instructions, including optional on-column DNAse digestion. The RNAs' concentration and purity were subsequently analyzed using NanoDrop™ 2000c spectro-photometer. The total RNA's integrity was determined using the Agilent Fragment Analyzer capillary gel electrophoresis system with the DNF-471 Standard Sensitivity RNA Analysis Kit. Reverse transcription was performed using PrimeScript™ RT Reagent Kit with gDNA Eraser (Takara Bio, USA) according to the manufacturer's protocol. In all, 1.0 µg of total RNA was used in each reverse transcription. Gene expression of target genes was determined via quantitative Real-Time PCR using a Bio-Rad CFX384 Real-Time PCR System. Each 15 µL reaction consisted of 7.5 µL Thermo Fisher Power Track SYBR Green Mastermix, 4.5 µL nuclease-free water, 0.375 µL forward primer, 0.375 µL reverse primer - each at a concentration of 10 µM, resulting in a final primer concentration of 0.25 µM in each reaction – and 1.5 µL cDNA sample (Primer sequences are listed in Supplementary Table 1). As thermal cycle protocol following parameters were used: 95 °C for 2 min followed by 40 cycles two-step PCR using 95 °C for 10 sec (denaturation) and 60 °C for 30 sec. (annealing and elongation) with a fluorescence detection step at the end of each cycle. Finally, a melting curve analyses was performed using the instrument's standard parameters. The expression intensity of each gene of interest (GOI) was determined using GAPDH, a highly expressed housekeeping gene showing no alterations in our RNA-Seq experiments, as a reference gene. A Ct-value of 35 was set as the technical reliability cutoff and treated as negative. The expressions were depicted as totally arbitrary expression units. PCR products were additionally controlled using a QIAGEN QIAxcel Advanced capillary gel electrophoresis system with the QIAGEN DNA Screening Kit.

## LEGENDplex™ assay

The protein concentration of IL-1α, IL-11 and TNF-α at time points between 6 and 54 h post mRNA transfection was quantified by LEGEN-Dplex™ assays (BioLegend, USA) from HeLa cell culture supernatants. In brief, cell debris-free cell culture supernatants were prepared for flow cytometry according to the manufacturer's protocol. Cytofluorometric analyses were performed with a flow cytometer FC500 and CXP analysis software (Beckman Coulter, USA). Data analysis was performed with the QOGNIT LEGENDplex™ Data Analysis Software Suite (version 2023-02-15).

## IL-24 ELISA

The protein concentration of IL-24 at time points between 6 h and 54 h post mRNA transfection was quantified using the Human IL-24 DuoSet ELISA Kit (Bio-Techne, USA). The assay was conducted according to the manu-facturer's protocol with cell debris-free cell culture supernatants. Mea-surement was carried out with a Multiscan FC photometer (Thermo Fisher Scientific, USA). Data was analyzed with GainData® (arigo Biolaboratories Corp., Taiwan).

## RNA-Seq and bioinformatics data analysis using pDNA constructs

HeLa cells were cultured and transfected using GFP-control and GFP-MYPOP expression plasmids. 24 h after transfection, HeLa cells were sorted for green fluorescence in order to exclude non-transfected cells from the analysis. Total RNA was isolated using the RNeasy Plus Micro Kit (QIAGEN, Netherlands). RNA was quantified with the Qubit™ RNA High Sensitivity (HS) Kit. RNA quality was assessed by on-chip electrophoreses with a Bioanalyzer 2100 (Agilent Technologies, USA) using an RNA Nano Chip. All samples showed RNA integrity numbers (RIN) above 9. RNA library preparation from 150 ng RNA input was performed with the NEBNext® Ultra™ II RNA library preparation kit for Illumina according to the manufacturer's protocol with a final amplification of 12 cycles. Quantity was assessed using the Qubit™ DNA High Sensitivity (HS) Kit, and quality was assessed by on-chip electrophoreses using Agilent's Bioanalyzer 2100 HS DNA assay. Sequencing was performed on an Illumina HiSeq 2500 (59 bp single-end reads).

Quality control on the sequencing data was performed with the FastQC tool (version 0.11.2, https://www.bioinformatics.babraham.ac.uk/projects/fastqc/). To quantify expression of viral genes derived from the inserted HPV genome, we merged both the Homo sapiens.GRCh38 reference genome and the annotation with HPV isolate 18 (NCBI identifier: NC_001357.1). RNA sequencing reads were then aligned to this reference. The corresponding annotation (ENSEMBL v107) was also retrieved from ENSEMBL FTP website. The STAR aligner (version 2.7.10a) was used to perform mapping to the reference genome[82]. Alignments were quantified with the featureCounts function of the Rsubread package (version 2.10.5[83]) against the ENSEMBL v107 annotation. Exploratory data analysis was performed with the pcaExplorer package (version 2.22.0[84]). For batch effect analysis the limma package (version 3.52.4[85]) and the sva package (version 3.44.0,[86]) were used. As a result, the batch of the samples was included in the different expression analysis model as well as the condition of the samples. Differential expression analysis was performed with edgeR package (version 3.38.4[87]), setting the false discovery rate (FDR) cutoff to 0.05. KEGG pathway enrichment was performed with the enrichr package (version 3.1[88]) and the KEGG_2019_human database. The enrichment results were further processed with the GeneTonic package for visualization and summarizing (version 2.0.2[89,90]). Gene expression profiles were plotted as heatmaps (color-coded standardized z-scores for the expression values, after variance stabilizing transformation using the DESeq2 package (version 1.36.0[91]) to simplify comparison across samples.

STRING analysis (version 11.5, https://string-db.org/) was conducted with all DEGs which were included in the KEGG Pathway 'cell cycle'. Minimum required interaction score was set to highest confidence (0.900).

## RNA-Seq and bioinformatics data analysis using mRNA constructs

Total RNA was extracted from dry frozen cell pellets using QIAGEN's RNeasy Mini Kit in case of samples with ≥1x10E6 cells or QIAGEN's RNeasy Micro Kit for samples with <1x10E6 cells. Quantity and quality of total RNA was assessed using the Qubit 4 fluorometer with the RNA BR AssayKit from Invitrogen and the Agilent Tapestation4200 system using the RNA ScreenTape Assay Kit. mRNA-focused RNA libraries were prepared using Illumina's TruSeq stranded RNA library prep kit with an input amount of 300 ng total RNA per sample according to the manufacturer's protocol with a final amplification of 12 cycles. Final library quantity and quality control was done using the Qubit 4 fluorometer with the dsDNA HS AssayKit from Invitrogen and the Agilent Tapestation4200 system using the D1000 ScreenTape Assay kit. All mRNA libraries were sequenced in paired-end mode (2 × 50 nt) on an Illumina NovaSeq 6000 instrument resulting in around 20 million distinct sequencing read pairs per library. Quality control on the sequencing data was performed with the FastQ Screen tool (0.15.1,[92]) and the FastQC tool (version 0.11.9,[93]). RNA sequencing reads were processed with kallisto (version 0.42.4,[94]) and the Homo sapiens reference GRCh38 (ENSEMBL release 86). Estimated read counts were summarized per gene and used for differential expression analysis with DESeq2 (version 1.34.0,[91]). Log2 fold change values after "apeglm" shrinkage[95] are shown. Genes with BH[96] adjusted p-values < 0.05 were considered as significantly regulated.

## Stable transduction of HELA cell line with Incucyte® cell cycle lentivirus reagent

The Incucyte® Cell Cycle Green/Orange Lentivirus Reagents (Sartorius, #4809) were used for stable labeling HELA cells for in vitro analysis of cell cycle progression. By linking fluorescent proteins TagGFP2 and TagRFP to fragments of Geminin and Cdt1, the G1 and S/G2/M phases can be monitored in real time[97]. As a result, cells in G1 appear predominantly red, cells in S/G2/M predominantly green, transition phases G1-S yellow and cells in M-G1 or non-transduced cells colorless. This color assignment forms the basis for the subsequent automatic classification of the cell cycle phases.

For transduction, 9500 HELA cells per well were seeded in a 24-well plate. Transduction itself was performed according to the manufacturers' protocol. The cells were transduced with an MOI of 3. Throughout the transduction process, the cells were monitored using the Incucyte® Live Cell Analysis System/Cell by Cell Analysis Software Module.

## Cell health assay of human MYPOP mRNA-transfected HELA cell line via Incucyte®

In all, 40,000 HELA cells per well were seeded in a 12-well plate and were incubated at 37 °C and 5% $CO_2$. After 24 h, cells were transfected with the respective mRNAs as described above. As further control, untreated cells were included. All samples were prepared in quadruplicate. Directly after transfection Incucyte® Annexin V NIR Dye (Sartorius, #4768) was added to each well, and the 12-well plate was placed in Incucyte® Live Cell Analysis System for visual monitoring over 72 h. Incucyte® Cell-by-Cell Analysis Software Module was used for analysis.

The Incucyte® Annexin V NIR Dye is a highly selective cyanine-based fluorescent dye ideally suited for a simple mix-and-read, real-time quantitative assay of apoptosis. This dye is non-perturbing to cell growth or morphology and yields little or no intrinsic fluorescent signal. Once cells become apoptotic, plasma membrane phosphatidylserine (PS) asymmetry is lost, leading to exposure of PS to the extracellular surface and binding of the Incucyte® Annexin V Dye, yielding a bright and photostable fluorescent signal. With the Incucyte® integrated analysis software, fluorescent objects can be quantified, and background fluorescence is minimized.

## Language optimization

The large language model ChatGPT (OpenAI, GPT-3.5) was used for language editing.

## Statistics and reproducibility

All experiments were performed in at least three biological replicates (*n*). Apart from RNA-Seq experiments, statistical data analysis was performed using GraphPad Prism 9 (version 9.5.0). A Gaussian distribution was assumed for all experiments. Accordingly, statistical significance was tested using a two-tailed unpaired *t* test with Welch's correction. In our hypothesis testing scenarios, we set alpha = 0.05. Statistically significant changes were marked in figures with * for $p \le 0.05$, ** for $p \le 0.01$, *** for $p \le 0.001$. Exact *p* values are listed in the respective figure legend.

## Reporting summary

Further information on research design is available in the Nature Portfolio Reporting Summary linked to this article.

## Data availability

Raw and processed files for RNA-Seq datasets generated in the scope of this manuscript using pDNA constructs for transfection are available in the Gene Expression Omnibus (GEO) database under the accession number GSE260896.

The data of the mRNA-mediated MYPOP gene transfer have been deposited in the European Nucleotide Archive (ENA) at EMBL-EBI under accession number PRJEB70469. Processed files for RNA-Seq datasets are provided in Supplementary Data 1. Numerical source data for the graphs in the manuscript can be found in Supplementary Data 2.

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

## Acknowledgements

The article contains essential parts of the doctoral theses of M.S., M.B.S., and S.W.

## Author contributions

J.S., S.M., T.B., K.R., S.K., A.K., M.D. and L.F. contributed to the conception or design of the work. J.S., A.Hü., M.S., T.D., M.P., M.B.S., S.W., K.F., M.K., G.H., F.M., A.L., C.O., A.He., B.S., A.N., M.S., S.B., M.P., A.K., M.D. and L.F. contributed to the acquisition, analysis, or interpretation of data. J.S., T.B., K.R., G.H., F.M., T.H., M.A.S., S.K., M.D. and L.F. provided funding and resources. J.S., A.Hü., M.S., M.P., S.W., S.M., M.K., G.H., F.M., A.L., T.H., C.O., M.A.S., A.He., B.S., A.N., M.S., S.K., A.K., M.D. and L.F. have drafted the manuscript or substantially revised it.

## Funding

J.S. discloses support for the research of this work from the Max Planck Graduate Center with the Johannes Gutenberg-Universität Mainz (MPGC). L.F. and F.M. disclose support for the research of this work from the Deutsche Forschungsgemeinschaft (DFG, German Research Foundation) [FL 696/6-1, 530277283] and [SFB1292/2 TP19N], respectively. M.A.S. received biobank funding from the German Center of Lung Research [82DZL00402]. This work has also been supported by the computing infrastructure provided by the Core Facility Bioinformatics at the University Medical Center Mainz. All other authors declare no relevant funding. Open Access funding enabled and organized by Projekt DEAL.

## Competing interests

The authors declare the following competing interests: L.F. and M.A.S. are inventors of the patent application related to this study. All other authors declare no competing interests. The funders had no role in study design, data collection and analysis, decision to publish, or preparation of the manuscript.

## Additional information

[1]Institute for Virology, University Medical Center of the Johannes Gutenberg-University Mainz, Mainz, Germany. [2]TRON - Translational Oncology at the University Medical Center of the Johannes Gutenberg University Mainz gGmbH, Mainz, Germany. [3]Institute of Immunology, University Medical Center of the Johannes Gutenberg-University Mainz, Mainz, Germany. [4]University Cancer Center Mainz, University Medical Center of the Johannes Gutenberg-University Mainz, Mainz, Germany. [5]German Cancer Consortium (DKTK), Heidelberg, Germany. [6]Research Center for Immunotherapy (FZI), University Medical Center of the Johannes Gutenberg-University Mainz, Mainz, Germany. [7]Cell Biology Unit, University Medical Center of the Johannes Gutenberg University Mainz, Mainz, Germany. [8]Wilkes University, Department of Biology, Wilkes Barre, PA, USA. [9]Institute of Medical Biostatistics, Epidemiology and Informatics (IMBEI), University Medical Center of the Johannes Gutenberg University Mainz, Mainz, Germany. [10]Institute of Organismic and Molecular Evolution, Molecular Genetics and Genome Analysis Group, Department of Biology, Johannes Gutenberg University Mainz, Mainz, Germany. [11]Translational Lung Research Center (TLRC) Heidelberg, German Center for Lung Research (DZL), Heidelberg, Germany. [12]Translational Research Unit (STF), Thoraxklinik at Heidelberg University Hospital, Heidelberg, Germany.
✉e-mail: lflorin@uni-mainz.de

