## [Transparent Peer Review file · Communications Biology]

The MYB-related transcription factor MYPOP acts as a selective regulator of cancer cell growth

Corresponding Author: Professor Luise Florin

Version 0:

Reviewer comments:

Reviewer #1

(Remarks to the Author)

In this manuscript, Strunk et al. investigate the phenotypic and cellular effects of the MYB-related Transcription Factor MYPOP on HeLa cells. Whilst the study provides some interesting data on the potential tumour suppressive functions of MYPOP, some of the data is unclear and not robust enough to support the author's claims. All of the data is based on the over-expression of MYPOP and this is insufficient due to potential issues with over-expressing a protein and non-physiological levels. Furthermore, all of the data is done in HeLa cells - thus, the ability to extrapolate their data to 'human cancers cells' is not justified and can only be claimed for HeLa cells. At this point, I find the work quite preliminary and several additional experiments are required to support the current data.

- In Fig 1a, what is the band below the purported MYPOP band in the pcDNA3.1-MYPOP transfected lane? Also, why is there two bands in the GFP-MYPOP transfected lane? What is the faint band in HeLa cells about 100kDa? These results and unexpected bands should be explained

- In Fig 1, the authors should provide a lower magnification image to show GFP-MYPOP localisation being both nuclear and cytoplasmic, as it looks almost completely nuclear in 1b and the data in 1c is not convincing as the cells are dying

- Is the data in Fig 1b from HeLa cells? This is not clear in the text or figure legend

- In Supp Fig 1, is the decrease in HPV18 E6 and E7 significant?

- In Fig 4, the data would be significantly strengthened by staining for a mitotic marker such as phosphorylated Ser10 on Histone H3. This should also be analysed by flow cytometry to strengthen the data

- The authors data solely relies on MYPOP over-expression in HeLa cells. Depletion of MYPOP in cancers cells should be performed to see if this enhances the transformed phenotype

- Wüstenhagen, E. et al. demonstrated that the HPV E7 protein is responsible for the decreased expression of MYPOP. The authors should deplete E7 in HeLa cells and see if this results in increased MYPOP expression, apoptosis and deregulated mitosis

- In Fig 8c, proper analysis of co-localisation between MYPOP and α -Tubulin, and MYPOP and condensed chromatin should be performed. Furthermore, the image provided for the localisation between MYPOP and condensed chromatin is unclear and seems to just show co-localisation with the nucleus

- All the studies in human cells is performed in one cell line, HeLa cells. To be able to extrapolate these data to human cancer cells more generally, additional cancer cell line should be used to ensure the effects are not just true for HeLa cells

Reviewer #2

(Remarks to the Author)

While Strunk et al.'s manuscript presents preliminary data suggesting a potential tumor-suppressing role for MYPOP,

additional experiments, and clarifications are needed to address significant gaps and inconsistencies.

The manuscript presents interesting data on the expression of MYPOP in cancer cell lines, such as HeLa. However, the data to support the hypothesized tumor-suppressing role of MYPOP through gene depletion or knockout studies is lacking. For a gene proposed as a tumor suppressor, it is crucial to demonstrate enhanced cellular transformation upon its loss. The authors should perform experiments where MYPOP is depleted or knocked out, followed by an analysis of the cellular outcomes to substantiate their claims.

There is a noticeable contradiction in detecting endogenous MYPOP between Fig 1A and Fig 5A (right panel). In Fig 1A, MYPOP is detectable in NHEK cells, whereas it is absent under similar conditions in Fig 5A. Clarification on experimental conditions or potential errors should be addressed to ensure the reproducibility and reliability of the results.

The manuscript does not address the presence of an additional band in HeLa cell lysates observed in some figures (e.g., Fig 5A).

The authors report that overexpression of MYPOP in HeLa cells leads to growth arrest. However, the underlying mechanisms remain unclear. For example, the repression of oncogenes such as MYC, E6, and E7 by MYPOP is mentioned but not mechanistically explored. Understanding whether MYPOP directly binds to regulate these oncogene transcripts or if it acts through modulation of other transcription factors or via epigenetic changes would significantly enhance the manuscript's impact.

Changes in the transcription profile attributed to MYPOP overexpression are suggested to be secondary effects. It would be good to delineate primary versus secondary transcriptional changes to elucidate this aspect further.

Version 1:

Reviewer comments:

Reviewer #1

(Remarks to the Author)

The authors have addressed all of my comments satisfactorily. My only remaining comment is that ref 57 is quite old and I suggest that a more up to date reference discussing how HPV can modulate cell cycle/genomic stability regulators may be more suitable (e.g. Scarth et al., J Gen Virol, 2021).

Reviewer #2

(Remarks to the Author)

This manuscript explores the role of the MYB-related transcription factor MYPOP (p42POP) as a selective regulator of tumor cell growth. Using a combination of plasmid- and mRNA-based gene delivery, transcriptomics, live-cell imaging, and functional assays across human and murine systems, the authors identify MYPOP as a tumor-specific growth suppressor that induces G1/S arrest, disrupts mitosis, silences oncogenes (including MYC and HPV E6/E7), and encourages cytokine release, especially IL-24.

The study investigates an understudied transcription factor and provides an extensive dataset indicating anti-proliferative effects of MYPOP across various cancer cell types while sparing normal cells. The range of experimental methods and consistency between DNA- and mRNA-based delivery are strengths. However, several conceptual, mechanistic, and presentation issues need to be addressed before the conclusions can be fully validated.

Major Comments

1. Although the data convincingly show growth inhibition and cell death caused by MYPOP overexpression, the manuscript remains largely descriptive. It is unclear whether MYPOP directly regulates important targets such as MYC, IL24, or cell-cycle genes, or whether these effects are indirect consequences of cell-cycle arrest or cellular stress. Including mechanistic details, such as chromatin binding, promoter occupancy, or transcriptional dependency analyses, would greatly improve the study.
2. While MYPOP knockout in transformed cells does not appear to affect growth, it would be essential to assess the impact of MYPOP loss in untransformed cells. Given that the knockout reagents are already available and validated, testing this would help clarify whether MYPOP functions as a bona fide tumor suppressor rather than exerting effects only upon overexpression.
3. Most conclusions are based on supra-physiological MYPOP expression achieved through plasmid or mRNA transfection. Although the authors show low endogenous MYPOP levels in cancer cells, it remains unclear whether restoring MYPOP to normal levels would produce similar effects. This distinction is important for understanding MYPOP as a genuine tumor suppressor rather than a factor that causes cytotoxicity through overexpression.
4. Transcriptomic analyses strongly suggest G1/S arrest, but flow cytometry-based DNA content analysis shows only minor or non-significant shifts. The FUCCI system provides supporting evidence, but the apparent discrepancies across cell cycle readouts should be discussed more clearly and reconciled.
5. The repression of MYC and HPV oncogenes (E6/E7) is clear at early time points but decreases by 24 hours despite ongoing MYPOP expression. This raises questions about feedback mechanisms and the stability of MYPOP-driven transcriptional control, particularly in therapeutic applications.
6. The selective effect on tumor versus normal cells is a key point. While NHEKs show minimal transcriptomic and phenotypic changes, the manuscript would benefit from validation across more normal primary cell types or a more precise explanation of why tumor cells are particularly vulnerable to MYPOP re-expression.

Minor Comments

1. Several sections, especially in the Results, are overly repetitive, restating similar conclusions across plasmid- and mRNA-based experiments. Streamlining would enhance readability without losing content.
2. Some conclusions are based on trends that do not reach statistical significance (e.g., E6/E7 repression at 24 h). These should be described more carefully in the text.
3. The manuscript switches between describing MYPOP as a “tumor suppressor” and a “selective growth inhibitor.” Given the reliance on overexpression systems, more cautious wording might be preferable.

Version 2:

Reviewer comments:

Reviewer #2

(Remarks to the Author)

The authors have adequately addressed the concerns raised in the previous review and have revised the manuscript accordingly. I am satisfied with the revisions and believe the manuscript is now suitable for publication.

Point by point response Strunk et al.

We are grateful for the referees' comments which were extremely helpful in guiding our revision.

Reviewer #1

- In Fig 1a, what is the band below the purported MYPOP band in the pcDNA3.1-MYPOP transfected lane? Also, why is there two bands in the GFP-MYPOP transfected lane? What is the faint band in HeLa cells about 100kDa? These results and unexpected bands should be explained

We thank the reviewer for this valuable comment. Further Western Blot analysis in HeLa cells (wild type versus MYPOP knockout) were performed and uncovered that the faint band at about 100kDa represents an unspecific band of the antibody as it is equally and well detectable in wild type (WT) and MYPOP-KO HeLa cells when more protein was loaded, while weak but specific MYPOP bands are detectable between 40 and 60 kDa (Supplementary Fig. 1) pointing towards post translational modified or cleaved forms of the protein. A new supplementary figure 1 as well as a paragraph (line 144-158) are included into the revised manuscript.

New Supplementary Fig. 1

- In Fig 1, the authors should provide a lower magnification image to show GFP-MYPOP localisation being both nuclear and cytoplasmic, as it looks almost completely nuclear in 1b and the data in 1c is not convincing as the cells are dying.

Fig.1b is now replaced by new Fig. 1b-f. Lower magnification images (b) showing nuclear and cytoplasmic localization for GFP-MYPOP expressing cells as well as close ups in c-e show different localizations. In addition, new Figure 1e (representative image) and 1f (co-localization quantification) demonstrate that nuclear GFP-MYPOP colocalizes with condensed chromatin (GFP-MYPOP/Hoechst: PCC +0.5; GFP/Hoechst: PCC -0,3) (reviewer point for Fig 8c, see below).

We also modified the main text (line 168/169) "... GFP-MYPOP showed predominantly nuclear localization in most cells, with additional nucleocytoplasmic and cytoplasmic distribution observed in some cases (Figs. 1b-e)."

New Fig. 1b-f

- Is the data in Fig 1b from HeLa cells? This is not clear in the text or figure legend

We now ensured that cells are specified in all subfigures (text and legend).

- In Supp Fig 1, is the decrease in HPV18 E6 and E7 significant?

We thank the reviewer for this point. Significance of E6/E7 mRNA levels were missing for both MYPOP DNA and mRNA transfection.

As the decrease after MYPOP DNA was not significant, we now added p-value to the figure legend of Supp Fig 2 (old Supplementary. Fig. 1) and modified the text (line 308): "We observed a downregulation of E6 and E7 transcripts to approximately 75% upon GFP-MYPOP overexpression in HeLa cells after 24 hours of DNA transfection; however, this reduction did not reach statistical significance".

As our qPCR data for E6 and E7 transcripts after MYPOP mRNA treatment did reach statistical significance for E6 transcripts, at 6h post treatment, we added the "***" to the figure Fig. 8a (old Fig. 7a) and toned down the statement that MYPOP silences viral oncogene expression throughout the manuscript.

Revised Fig. 8a (old Fig. 7a)

- In Fig 4, the data would be significantly strengthened by staining for a mitotic marker such as phosphorylated Ser10 on Histone H3. This should also be analysed by flow cytometry to strengthen the data

We are grateful for this point and reanalyzed cell cycle by using live cell imaging (Incucyte technology) in HeLa cells stably expressing the Fucci reporters hCdt1-RFP (G1 marker, red) and hGeminin-GFP2 (S/G2/M marker, green) to monitor cell cycle progression in real-time (Fucci sensors: powerful new tools for analysis of cell proliferation, J Biol chem. 2001). Here, we got strong indication that MYPOP blocks G1/S transition. These new results are in line with our RNA seq results and are now incorporated as new paragraph "MYPOP-induced G1/S arrest" (line: 667- 686) and new Fig.9. Old Figs. 4a-d and 8a,b using flow cytometric analysis and Hoechst as a DNA stain to analyze cell cycle were removed from the manuscript and replaced by new Fig. 9. (old Fig.8 c,d,e,f are now shown as Supplementary Fig. 6a-c)

New Fig. 9

Old Fig. 8c-f is now shown as new Supplementary Fig. 6 a-d.

- The authors data solely relies on MYPOP over-expression in HeLa cells. Depletion of MYPOP in cancers cells should be performed to see if this enhances the transformed phenotype

Du et al. previously reported that MYPOP depletion using siRNA promoted cell elongation, as well as increased cell migration and survival in various cell lines (e.g., HEK293T, HaCaT, and MCF-7).

To directly address the reviewer's concern, we performed a CRISPR-mediated knockout of MYPOP in HeLa cells. In contrast to the findings of Du et al., we did not observe any changes in cell morphology or growth behavior (new Supplementary Fig. 1 – see above and new Fig. 6c). Furthermore, MYPOP siRNA treatment in 2106T cells did not affect cell growth (new Supplementary Fig. 4).

New Fig. 6c

new Supplementary Fig. 4

In light of these results, we also incorporated a discussion of the findings by Du et al.,: (line 948-954)“Interestingly, although Du et al. convincingly reported increased migration and survival after MYPOP siRNA treatment of HEK293T, HaCaT, and MCF-7 cells, our finding that MYPOP depletion does not affect HeLa or 2106T morphology or growth indicates that MYPOP’s influence is context-dependent as reported for multiple TFs⁴. These findings also suggest that MYPOP expression in HeLa and 2106T cells is already repressed below the threshold necessary for growth control, and that further reduction does not confer any additional proliferative advantage as reported for various tumor suppressors⁷⁵.” In addition, we have softened our claim that MYPOP functions as a tumor suppressor throw-out the manuscript.

- Wüstenhagen, E. et al. demonstrated that the HPV E7 protein is responsible for the decreased expression of MYPOP. The authors should deplete E7 in HeLa cells and see if this results in increased MYPOP expression, apoptosis and deregulated mitosis

The depletion of HPV E7 and its effects on apoptosis and mitotic deregulation have already been extensively investigated in previous studies. For example, Butz et al. (Oncogene, 2003) and Yamato et al. (Cancer Gene Ther., 2008) demonstrated that siRNA-mediated silencing of HPV oncogenes E6 and E7 efficiently induces cell death in HPV-positive cervical cancer cells. In our study, we analyzed the effect of E7 knockdown as well as proteasome inhibitors on MYPOP expression. However, treatment of HPV-positive cells with E7 siRNA or inhibitors led to extensive cell death before a measurable restoration of MYPOP protein levels could be observed. Given these limitations, a more detailed investigation of the temporal relationship between E7 depletion, MYPOP re-expression, and cellular outcomes is an interesting avenue for future research.

- In Fig 8c, proper analysis of co-localisation between MYPOP and α -Tubulin, and MYPOP and condensed chromatin should be performed. Furthermore, the image provided for the localisation between MYPOP and condensed chromatin is unclear and seems to just show co-localisation with the nucleus

We thank the reviewer for this comment. We now quantified co-localization using the Pearson’s correlation coefficient (PCC). New Figures 1e and f are added showing (representative image) and 1f (co-localization quantification) demonstrating that nuclear GFP-MYPOP colocalizes with condensed chromatin (GFP-MYPOP/Hoechst: PCC +0.5; GFP/Hoechst: PCC -0,2). In contrast, proper analysis of co-localization between MYPOP

and α -tubulin demonstrated that there is no linear correlation suggesting likely independent distribution between MYPOP and tubulin ($PCC=0$). Therefore, we removed the statement from the manuscript.

New Fig. 1e-f

- All the studies in human cells is performed in one cell line, HeLa cells. To be able to extrapolate these data to human cancer cells more generally, additional cancer cell line should be used to ensure the effects are not just true for HeLa cells

We completely agree with this point and performed a series of experiments using MYPOP pDNA or mRNA. The new figure 5 demonstrates that MYPOP expression is broadly diminished across various human cancer types and that its restoration exerts a pronounced growth-suppressive effect (new Figure 5 for pDNA and revised figure 6 (old Fig.5) for mRNA transfection.

New Paragraphs line 426-438 "MYPOP expression reduces tumor cells growth" and 489-503 are included into the revised manuscript.

New Fig. 5 MYPOP pDNA transfection reduces tumor cells growth.

New Fig. 6: MYPOP expression modulates cell growth of tumor cells.

Reviewer #2:

The manuscript presents interesting data on the expression of MYPOP in cancer cell lines, such as HeLa. However, the data to support the hypothesized tumor-suppressing role of MYPOP through gene depletion or knockout studies is lacking. For a gene proposed as a tumor suppressor, it is crucial to demonstrate enhanced cellular transformation upon its loss. The authors should perform experiments where MYPOP is depleted or knocked out, followed by an analysis of the cellular outcomes to substantiate their claims.

Ectopic expression of MYPOP markedly suppressed tumor cell proliferation, consistent with a growth-inhibitory function. In contrast, knockout in HeLa cells (New Fig. 6c) or siRNA-mediated depletion in 2106T cells (New Supplementary Fig. 4) of the already low endogenous MYPOP levels had no measurable effect on cell growth of HeLa or 2106T cells. This observation suggests that MYPOP expression in tumor cells is already repressed below the threshold necessary for growth regulation, and that further reduction does not confer additional proliferative advantages—similar to what has been observed for other tumor suppressors such as p53 (Lenoir, W. F. et al. Nat. Commun. 12, 6506, 2021).

New Fig. 6c

new Supplementary Fig. 4

In contrast, Du et al., found effects of MYPOP siRNA treatment on HaCaT, MCF-7 and HEK293T cells on survival and migration as mentioned in the introduction and now in the discussion (line 948-954) “Interestingly, although Du et al. convincingly reported increased migration and survival after MYPOP siRNA treatment of HEK293T, HaCaT, and MCF-7 cells, our finding that MYPOP depletion does not affect HeLa or 2106T morphology or growth indicates that MYPOP’s influence is context-dependent as reported for multiple TFs⁴. These findings also suggest that MYPOP expression in HeLa and 2106T cells is already repressed below the threshold necessary for growth control, and that further reduction does not confer any additional proliferative advantage as reported for various tumor suppressors⁷⁵.” In addition, we have softened our claim that MYPOP functions as a tumor suppressor throughout the manuscript.

1. There is a noticeable contradiction in detecting endogenous MYPOP between Fig 1A and Fig 5A (right panel). In Fig 1A, MYPOP is detectable in NHEK cells, whereas it is absent under similar conditions in Fig 5A. Clarification on experimental conditions or potential errors should be addressed to ensure the reproducibility and reliability of the results.

We apologize for the earlier lack of clarity. Explanations have now been added to the main text and the legend. Main text (line 464-478): “Endogenous MYPOP was not detected in NHEK cells at 6 h p.t. due to low protein loading and short exposure times optimized to prevent oversaturation of overexpressed MYPOP. In HeLa cells, mRNA-induced MYPOP expression decreased but remained elevated, with additional bands below 55 kDa likely representing unmodified or cleaved forms.” Legend (538-545): “Under conditions of low protein loading and short exposure times, endogenous MYPOP is not detected in NHEK, as these parameters are optimized to prevent oversaturation of overexpressed MYPOP at 6 h p.t. Detection of endogenous MYPOP increases at later time points, reflecting cell growth and higher total protein content.”

2. The manuscript does not address the presence of an additional band in HeLa cell lysates observed in some figures (e.g., Fig 5A).

We thank the reviewer for this valuable comment. Further Western Blot analysis in HeLa cells (wild type versus MYPOP knockout) were performed for additional clarification and

uncovered that the faint band at about 100kDa represents an unspecific band of the antibody as it is equally and well detectable in wild type (WT) and MYPOP-KO HeLa cells when more protein was loaded, while weak but specific MYPOP bands are detectable between 40 and 60 kDa (Supplementary Fig. 1) pointing towards post translational modified or cleaved forms of the protein. A new supplementary figure 1 as well as a paragraph (line 144-158) are included into the revised manuscript.

New Supplementary Fig. 1

3. The authors report that overexpression of MYPOP in HeLa cells leads to growth arrest. However, the underlying mechanisms remain unclear. For example, the repression of oncogenes such as MYC, E6, and E7 by MYPOP is mentioned but not mechanistically explored. Understanding whether MYPOP directly binds to regulate these oncogene transcripts or if it acts through modulation of other transcription factors or via epigenetic changes would significantly enhance the manuscript's impact.

We thank the reviewer for this thoughtful comment and fully agree that elucidating the mechanistic basis of MYPOP-induced growth arrest is an important aspect. Previous studies have already provided relevant insights, and we have now expanded the Discussion to contextualize our findings within these mechanistic frameworks. The revised text reads as follows:

“Transient MYPOP expression resulted in downregulation of the oncogenic targets MYC, HPV E6, and E7, consistent with earlier reports demonstrating MYPOP binding to the HPV promoter as well as to MYB-consensus sites present and functional in the MYC promoter (Wüstenhagen et al., 2018; Lederer et al., 2005; Berge et al., 2007; Nakagoshi et al., 1992).

These data suggest that MYPOP can directly modulate transcription of oncogenic targets. Notably, although E6/E7 transcript levels decreased to ~30% at 6 h, repression was not sustained despite persistent MYPOP mRNA and protein expression, indicating possible compensatory feedback mechanisms, context-dependent promoter occupancy, or post-translational modification of MYPOP.”

We agree that the precise mechanistic pathways—whether through direct DNA binding, modulation of other transcription factors, or epigenetic mechanisms—remain complex and merit systematic investigation in future studies. Nevertheless, in the current manuscript we now provide additional mechanistic insight: our transcriptomic data demonstrate that MYPOP downregulates cell-cycle regulators, particularly G1/S drivers, and our newly added live-cell analyses confirm a G1/S arrest followed by apoptosis (new Fig. 9). Together, these results strengthen the connection between MYPOP overexpression and cell-cycle control, offering a clearer mechanistic basis for the observed growth arrest.

New Fig. 9

4. Changes in the transcription profile attributed to MYPOP overexpression are suggested to be secondary effects. It would be good to delineate primary versus secondary transcriptional changes to elucidate this aspect further.

We thank the reviewer for this valuable comment and apologize for any misunderstanding.

In the revised manuscript, we now clarify that the transcriptional changes observed upon MYPOP overexpression occur early after MYPOP gene delivery, suggesting that these represent primary effects.

However, we acknowledge that both primary and secondary transcriptional changes likely contribute to the overall response. For example, the oncogenes E6, E7, and MYC are downregulated as early as 6 hours post-mRNA treatment, with this effect diminishing over time. This rapid downregulation, together with the immediate reduction in cell cycle gene expression and the observed G1/S arrest, supports a primary effect of MYPOP. In contrast, subsequent induction of cell death occurs later and is therefore interpreted as a secondary effect. This and additional aspects related to these mechanisms are now discussed in the revised manuscript.

We added e.g. in Line 996-1000: "Early effects likely involve downregulation of G1/S-promoting genes, including oncogenes. These primary transcriptional changes may drive the observed secondary effects on induction of the tumor suppressor IL-24, microtubule integrity, collectively contributing to MYPOP's anti-proliferative and death inducing effects observed in various tumor cells." As well as: (line 1001-1003): "Future studies should delineate MYPOP's direct genomic targets to distinguish between primary and secondary effects on gene regulation and to assess their respective contributions to cancer cell death."

We fully agree that further investigation of MYPOP's direct DNA-binding capacity (as shown for the HPV oncogene promoter or the Myb-binding site on DNA (as published by Wüstenhagen 2018 and Lederer 2005) and its potential interactions with chromatin or other transcriptional regulators will be crucial to fully elucidate its mechanism of action.

Accordingly, we added a new paragraph to the Discussion section (line 831-869).

Further revised figures:

Revised Fig. 10: we observed a mistake in Fig.10 statistical significance. Accordingly, we removed significance "*" from CT-26 cell count analysis and changed CT26 and B16-F10 position.

Point-by-point response to reviewers:

We thank the reviewers for their valuable and constructive comments, which were instrumental in improving our work.

Reviewers' comments:

Reviewer #1:

The authors have addressed all of my comments satisfactorily. My only remaining comment is that ref 57 is quite old and I suggest that a more up to date reference discussing how HPV can modulate cell cycle/genomic stability regulators may be more suitable (e.g. Scarth et al., J Gen Virol, 2021).

We thank Reviewer #1 for the positive evaluation of our manuscript. We agree that Scarth et al. provides a more up-to-date and appropriate reference regarding HPV-induced modulation of the cell cycle and genomic stability. This reference has now been cited in the Results and Discussion sections as Ref. 43.

Reviewer #2 (Remarks to the Author):

This manuscript explores the role of the MYB-related transcription factor MYPOP (p42POP) as a selective regulator of tumor cell growth. Using a combination of plasmid- and mRNA-based gene delivery, transcriptomics, live-cell imaging, and functional assays across human and murine systems, the authors identify MYPOP as a tumor-specific growth suppressor that induces G1/S arrest, disrupts mitosis, silences oncogenes (including MYC and HPV E6/E7), and encourages cytokine release, especially IL-24.

The study investigates an understudied transcription factor and provides an extensive dataset indicating anti-proliferative effects of MYPOP across various cancer cell types while sparing normal cells. The range of experimental methods and consistency between DNA- and mRNA-based delivery are strengths.

We thank the reviewer for this positive assessment.

However, several conceptual, mechanistic, and presentation issues need to be addressed before the conclusions can be fully validated.

In response to this comment, we have softened our conclusions and included a clear discussion of the study's limitations. Please see below for details.

Major Comments

1. Although the data convincingly show growth inhibition and cell death caused by MYPOP overexpression, the manuscript remains largely descriptive. It is unclear whether MYPOP directly regulates important targets such as MYC, IL24, or cell-cycle genes, or whether these effects are indirect consequences of cell-cycle arrest or cellular stress. Including mechanistic details, such as chromatin binding, promoter occupancy, or transcriptional dependency analyses, would greatly improve the study.

We agree with the reviewers point and therefore wrote already in the first point-by-point letter: "We fully agree that further investigation of MYPOP's direct DNA-binding

capacity (as shown for the HPV oncogene promoter or the Myb-binding site on DNA and as published by Wüstenhagen 2018 and Lederer 2005) and its potential interactions with chromatin or other transcriptional regulators will be crucial to fully elucidate its mechanism of action.

Accordingly, to highlight the limitations of the study, we now wrote more carefully and modified the discussion (line 615): “One of the key findings of this study is MYPOPs impact on gene expression. Transient MYPOP expression caused downregulation of oncogenic targets such as MYC and HPV E6, consistent with earlier evidence for MYPOP binding to the MYB-consensus sites which are present and functional in both the MYC and the HPV promoter 8,10,53,54. At the moment, we can just speculate that MYPOP directly modulates transcription of these oncogenic targets. Although E6/E7 transcript levels decreased to ~30% after 6h, repression was not sustained despite persistent MYPOP mRNA and protein, suggesting compensatory feedback, context-dependent modulation of promoter occupancy or MYPOP modification. Detection of MYPOP protein species at ~60 kDa and lower molecular weights points to differential modification or proteolytic processing, potentially limiting activity of MYPOP protein and thus influencing chromatin binding and transcriptional efficacy. Mapping of DNA binding sites in diverse tumor and normal cells, would greatly improve our understanding on how MYPOP acts mechanistically.”

In line 582, we removed:” restored transcriptional control over key oncogenes”.

2. While MYPOP knockout in transformed cells does not appear to affect growth, it would be essential to assess the impact of MYPOP loss in untransformed cells. Given that the knockout reagents are already available and validated, testing this would help clarify whether MYPOP functions as a bona fide tumor suppressor rather than exerting effects only upon overexpression.

We agree with the reviewer that assessing MYPOP loss-of-function in untransformed cells would be important to further clarify its role as a bona fide tumor suppressor.

We therefore acknowledge that a more comprehensive loss-of-function analysis in normal cells, ideally using alternative genetic approaches, will be required to fully elucidate the role of MYPOP in non-transformed cellular contexts. This limitation has now been explicitly stated in the Discussion.

Line 734: “The preferential effect of MYPOP re-expression on tumor cells remains unresolved. As our and the above-mentioned study are based on MYPOP overexpression, or loss-of-function analyses in tumor cells, efficient MYPOP knockout analyses in primary untransformed cells, will be important to determine whether MYPOP acts as a bona fide tumor suppressor under physiological conditions.”

3. Most conclusions are based on supra-physiological MYPOP expression achieved through plasmid or mRNA transfection. Although the authors show low endogenous MYPOP levels in cancer cells, it remains unclear whether restoring MYPOP to normal levels would produce similar effects. This distinction is important for understanding MYPOP as a genuine tumor suppressor rather than a factor that causes cytotoxicity through overexpression.

We appreciate the reviewer’s concern regarding the use of supra-physiological MYPOP expression levels. We now explained better (line 293): “In our single-cell, imaging-based analyses, MYPOP expression varied widely between individual cells, ranging from very low to very high levels. Notably, across all MYPOP-GFP-expressing cells, including those with weak expression, we did not detect cells at 24h p.t.

undergoing mitosis, with only a single cell showing metaphase chromosomes at 48h p.t. in which inhibition of sister chromatid separation could not be excluded (Fig. 4a).” and (line 491): “Mitotic activity was nearly eliminated by 24h independent of the observed expression level, accompanied by widespread microtubule disorganization. A slight recovery at 48 h p.t. suggests that MYPOP induces a transient mitotic block that recovers over time and is not strictly dependent on expression level.”

This is further supported by the partial relief of G1/S arrest observed at 24 h post mRNA transfection.

More importantly, supra-physiological MYPOP expression did not affect proliferation of normal keratinocytes, indicating that MYPOP overexpression is not intrinsically cytotoxic. Consistent with this context-dependent effect, cancer cell lines such as HCT116 showed an initial reduction in cell numbers followed by full recovery within three days (Fig. 6b), whereas HeLa and 2106T cells exhibited no increase in cell numbers over the same period.

Together, these data indicate that MYPOP induces a transient, reversible mitotic arrest across a range of expression levels and acts in a context-dependent manner rather than being a nonspecific cytotoxic factor. We agree that further experiments are needed to support MYPOPs role as a tumor suppressor.

The discussion is now modified accordingly: Line 676: “This resulted in a marked reduction in cell numbers in most tested human and mouse tumor cell lines, a transient growth inhibition in cells such as HCT116, and undetectable effects on normal keratinocytes, indicating that MYPOP overexpression acts in a context-dependent manner rather than as a nonspecific cytotoxic factor.”

4. Transcriptomic analyses strongly suggest G1/S arrest, but flow cytometry-based DNA content analysis shows only minor or non-significant shifts. The FUCCI system provides supporting evidence, but the apparent discrepancies across cell cycle readouts should be discussed more clearly and reconciled.

In the first round of revision, Reviewer #1 recommended using cell cycle markers instead of Hoechst staining and flow cytometry. Accordingly, we removed the flow cytometry–based DNA content analysis, which was performed at a single time point, and replaced it with live-cell Incucyte analysis using HeLa cells expressing FUCCI G1/G2 reporters. This state-of-the-art approach provides improved temporal resolution and supports a transient G1/S arrest immediately after MYPOP mRNA treatment, with recovery after 24 hours. The removed flow cytometry data are therefore no longer discussed in the manuscript.

5. The repression of MYC and HPV oncogenes (E6/E7) is clear at early time points but decreases by 24 hours despite ongoing MYPOP expression. This raises questions about feedback mechanisms and the stability of MYPOP-driven transcriptional control, particularly in therapeutic applications.

We agree with the reviewer that the reduced repression of MYC and HPV oncogenes at 24 hours despite continued MYPOP expression suggests the involvement of adaptive or feedback mechanisms.

We have therefore weakened our conclusions and explicitly discuss the possibility of feedback regulation and limited temporal stability/activity of MYPOP protein and

MYPOP-driven transcriptional control, particularly in the context of therapeutic applications.

Discussion, section line 620: “Although E6/E7 transcript levels decreased to ~30% after 6h, repression was not sustained despite persistent MYPOP mRNA and protein, suggesting compensatory feedback, context-dependent modulation of promoter occupancy or MYPOP modification. Detection of MYPOP protein species at ~60 kDa and lower molecular weights points to differential modification or proteolytic processing, potentially limiting activity of MYPOP protein and thus influencing chromatin binding and transcriptional efficacy. Mapping of DNA binding sites in diverse tumor and normal cells, would greatly improve our understanding on how MYPOP acts mechanistically.”

Line 752: “Collectively, these findings suggest that MYPOP-based gene delivery may have transient and context-dependent anti-tumor effects. Whether sustained delivery or combination treatments can meaningfully enhance its activity, similar to other G1/S-blocking agents 77, remains to be determined. Substantial preclinical in vivo studies are still needed to assess its safety and overall therapeutic potential.”

6. The selective effect on tumor versus normal cells is a key point. While NHEKs show minimal transcriptomic and phenotypic changes, the manuscript would benefit from validation across more normal primary cell types or a more precise explanation of why tumor cells are particularly vulnerable to MYPOP re-expression.

Line 691: “This differential responsiveness may arise from cell-type-specific differences in chromatin landscape or co-factor availability at MYPOP target promoters and warrants further investigation including validation across more normal primary cell types. Previous studies have shown that cooperative or antagonistic interactions among transcription factors can profoundly influence their occupancy at regulatory DNA elements 4 . Thus, delineating these TF interactions will be critical for understanding the context-dependent functions of MYPOP. In this scenario, the endogenous protein levels of MYPOP might play a role. Endogenous MYPOP is readily detectable in primary keratinocytes but strongly reduced in cancer cells as shown here and previously 8, suggesting that normal cells are intrinsically adapted to tolerate MYPOP activity, whereas tumour cells have undergone selection to downregulate it and thus become dependent on its loss. Consequently, re-expression of MYPOP in tumour cells may unmask this acquired vulnerability and elicits anti-proliferative and pro-apoptotic responses, while even supra-physiological MYPOP levels in NHEKs cause only minor transcriptomic and phenotypic changes.”

Line 749: “Additional studies are needed to identify MYPOP’s direct genomic targets and to explain its context-dependent effects...”

Minor Comments

1. Several sections, especially in the Results, are overly repetitive, restating similar conclusions across plasmid- and mRNA-based experiments. Streamlining would enhance readability without losing content.

We agree and have therefore moved counting of mitotic cells, disturbed microtubules, and respective immunofluorescences, as well as apoptosis assays after mRNA treatment to the supplementary part (Supplementary Fig. 6).

2. Some conclusions are based on trends that do not reach statistical significance (e.g., E6/E7 repression at 24 h). These should be described more carefully in the text.

This point was already raised by reviewer #1 in the previous round of revision and we either described more carefully.

We answered to reviewer #1 and modified the text (line 269): “We observed a downregulation of E6 and E7 transcripts to approximately 75% upon GFP-MYPOP overexpression in HeLa cells after 24 hours of DNA transfection; however, this reduction did not reach statistical significance”. In addition, we toned down the statement that MYPOP silences viral oncogene expression throughout the manuscript.

In addition, we discussed the putative compensatory feedback. Please see above.

3. The manuscript switches between describing MYPOP as a “tumor suppressor” and a “selective growth inhibitor.” Given the reliance on overexpression systems, more cautious wording might be preferable.

We agree with the reviewer’s point. Accordingly, we have further softened our conclusions and harmonized the terminology throughout the manuscript to describe MYPOP as a growth inhibitor/suppressor.

As MYPOP has been suggested as tumor suppressor, we only used this expression to describe previous work: “...and proposed tumor suppressor.”

And at the end of discussion (line 734) “The preferential effect of MYPOP re-expression on tumor cells remains unresolved. As our and the above-mentioned study are based on MYPOP overexpression, or loss-of-function analyses in tumor cells, efficient MYPOP knockout analyses in primary untransformed cells, will be important to determine whether MYPOP acts as a bona fide tumor suppressor under physiological conditions”.

We also rewrote the Conclusions section and used a more cautious wording.

Point-by-point response to reviewers:

Reviewers' comments:

Reviewer #1:

The authors have addressed all of my comments satisfactorily.

Reviewer #2:

The authors have adequately addressed the concerns raised in the previous review and have revised the manuscript accordingly. I am satisfied with the revisions and believe the manuscript is now suitable for publication.

We thank the reviewers for their valuable and constructive comments, which were instrumental in improving our work and now for the positive evaluation of our manuscript.